# Advancing Video Anomaly Detection:
# A Concise Review and a New Dataset

**Liyun Zhu**[1]     **Lei Wang**[1,2,*]     **Arjun Raj**[1]     **Tom Gedeon**[3]     **Chen Chen**[4]

[1]Australian National University, [2]Data61/CSIRO,
[3]Curtin University, [4]University of Central Florida
{liyun.zhu, lei.w, u7526852}@anu.edu.au,
tom.gedeon@curtin.edu.au, chen.chen@crcv.ucf.edu

## Abstract

Video Anomaly Detection (VAD) finds widespread applications in security surveillance, traffic monitoring, industrial monitoring, and healthcare. Despite extensive research efforts, there remains a lack of concise reviews that provide insightful guidance for researchers. Such reviews would serve as quick references to grasp current challenges, research trends, and future directions. In this paper, we present such a review, examining models and datasets from various perspectives. We emphasize the critical relationship between model and dataset, where the quality and diversity of datasets profoundly influence model performance, and dataset development adapts to the evolving needs of emerging approaches. Our review identifies practical issues, including the absence of comprehensive datasets with diverse scenarios.[2] To address this, we introduce a new dataset, Multi-Scenario Anomaly Detection (MSAD), comprising 14 distinct scenarios captured from various camera views. Our dataset has diverse motion patterns and challenging variations, such as different lighting and weather conditions, providing a robust foundation for training superior models. We conduct an in-depth analysis of recent representative models using MSAD and highlight its potential in addressing the challenges of detecting anomalies across diverse and evolving surveillance scenarios. [**Project website**]

## 1   Introduction

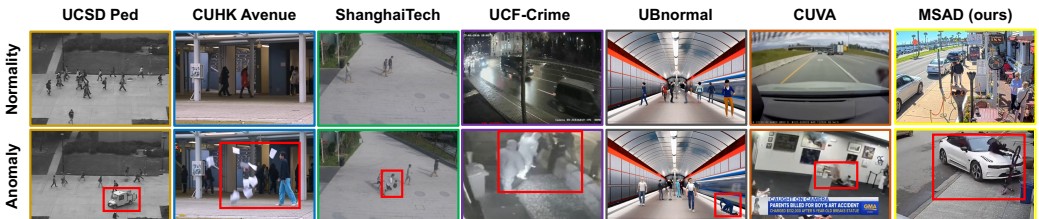

Figure 1: A comparison of existing datasets such as UCSD Ped, CUHK Avenue, ShanghaiTech, UCF-Crime, UBnormal and CUVA *vs.* our Multi-Scenario Anomaly Detection (MSAD) dataset.

Video Anomaly Detection (VAD) aims to automatically identify unusual occurrences in videos, enabling various applications in surveillance and monitoring [99]. Detecting anomalies is a challenging

---

*Corresponding author.

[2]In this paper, the term 'scenario' is used to signify different contexts such as retail, manufacturing, the education sector, smart cities, and others. In the context of different camera views, existing literature sometimes uses the term 'scene' to refer to the camera scene from a specific camera viewpoint.

38th Conference on Neural Information Processing Systems (NeurIPS 2024) Track on Datasets and Benchmarks.

and complex task due to several factors: (i) There is no unified and clear definition of anomalies of interest.[3] For example, the distinction between normal activities like walking on a sidewalk and abnormal activities like walking on a highway is context-dependent. (ii) The sporadic and rare occurrences of anomalies make the collection of well-curated datasets a demanding task, limiting the ability to learn anomalous patterns. Existing methods often treat VAD as either a one-class classification problem or an out-of-distribution detection task [4, 26, 46, 7, 30, 124, 24, 73, 33, 95, 61]. These approaches rely on training solely with normal samples and testing on both normal and abnormal samples, treating anomalies as outliers. Significant progress has been made in VAD during the past few years, thanks to benchmark datasets [45, 93, 47, 85, 18, 123], as well as synthetic data generation [2] that presents a range of anomalies. Fig. 1 shows some frames from different datasets.

**Surveys.** Advances in VAD have also led to some comprehensive surveys contributing significantly to the literature [18, 84, 60, 1, 81, 34, 115, 56]. A survey on deep learning models for VAD can be found in [84]. A comprehensive study in [1] highlights challenging issues in VAD, such as varying environments, the complexity of human activities, the ambiguous nature of anomalies, and the absence of appropriate datasets. A systematic review in [81] discusses various potential challenges, opportunities, and provides guidance for future research. A recent work [18] offers a comprehensive benchmark for understanding VAD causation using Large Language Models (LLMs) for text annotations on human-related anomalies. While we focus on static cameras, a survey on VAD in dynamic scenes captured by moving cameras can be found in [34]. Although these surveys are comprehensive, they are not portable and lightweight. Our lightweight review offers several benefits: (i) it provides a quick reference for researchers and practitioners, making it easier to get up to speed on VAD without sifting through extensive details; (ii) it enhances accessibility for a broader audience, including newcomers to the field, by allowing a quick grasp of essential concepts; and (iii) it offers focused guidance by distilling critical information into clear, actionable insights.

**Challenges.** Various studies [130, 89] indicate that existing methods are often restricted to detecting only a handful of specific anomalies due to (i) a limited amount of videos and (ii) limited camera viewpoints, scenarios and anomaly types per dataset. These methods also frequently suffer from poor generalizability, necessitating retraining for each unique target camera viewpoint or new scenario, which subsequently increases computational costs [11]. Most existing methods [71, 19, 10, 123] are particularly vulnerable to various factors such as reflection, illumination changes, and complex background environments, leading to frequent false positives and negatives, thereby affecting the precision and reliability of detection. These challenges highlight the need for a high-quality, multi-scenario, and comprehensive dataset. However, existing benchmarks mostly focus on single-scenario (either single or multiple camera viewpoints), human-related anomalies. No existing works explore the multi-scenario generalization abilities of VAD from a practical perspective.

**Motivations & contributions.** Given the aforementioned challenges in VAD, along with practical problems often ignored by the community, we are motivated to compile this lightweight and insightful survey to bridge this gap and provide valuable insights to both interested readers and domain experts. Our survey begins with a discussion of existing VAD methods (Sec. 2.1), highlights the critical relationship between model and dataset development, as well as recent trends and potentials. Our concise review offers fruitful insights between models and datasets (Sec. 2.2), motivating us to contribute a new dataset (Sec. 3), Multi-Scenario Anomaly Detection (MSAD), to advance VAD. We conduct a deep analysis of recent representative methods using our dataset and demonstrate its potential for addressing the challenges of detecting anomalies across diverse and evolving surveillance scenarios (Sec. 4). Finally, we conclude our work (Sec. 5).

## 2 A Concise Review

Our aim is to offer a lightweight reference for researchers and practitioners striving to advance VAD. Below we show the relationship between model and dataset development via a review on VAD

---

[3]Although there is no unified and clear definition of specific anomalies. Generally, we improve the definition of anomaly in the context of video anomaly detection, an anomaly refers to any event, behavior, or object in a video sequence that deviates from the normal or expected pattern of events. Given that anomalies are highly scene-dependent, single-scene datasets have a distinct advantage in testing a model's performance under consistent, controlled conditions. This allows for a more focused evaluation of the model's ability to detect deviations within a specific context.

Table 1: Comparisons between our Multi-Scenario Anomaly Detection (MSAD) dataset and existing datasets. Our MSAD is the first large-scale, comprehensive benchmark for real-world multi-scenario video anomaly detection. It has 35 human-related anomalies (HRA) and 20 non-human-related anomalies (NHRA). Our dataset (i) comprises 14 distinct scenarios, including roads, malls, parks, sidewalks, and more (ii) incorporates various objects like pedestrians, cars, trunks, and trains, along with (iii) dynamic environmental factors such as different lighting and weather conditions.

| Dataset | Year | Source | Domain | #Video | #HRA | #NHRA | #View | #Scenario | Modality | Resolution | Variations |
|---|---|---|---|---|---|---|---|---|---|---|---|
| Subway Entrance [3] | 2008 | Surveillance | Pedestrian | 1 | 5 | - | 1 | 1 | RGB | 512×384 | ✗ |
| Subway Exit [3] | 2008 | Surveillance | Pedestrian | 1 | 3 | - | 1 | 1 | RGB | 512×384 | ✗ |
| UMN [54] | 2009 | Surveillance | Behavior | 5 | 1 | - | 3 | 1 | RGB | 320×240 | ✗ |
| UCSD Ped1 [93] | 2010 | Surveillance | Pedestrian | 70 | 5 | - | 1 | 1 | RGB | 238×158 | ✗ |
| UCSD Ped2 [93] | 2010 | Surveillance | Pedestrian | 28 | 5 | - | 1 | 1 | RGB | 238×158 | ✗ |
| CUHK Avenue [45] | 2013 | Surveillance | Pedestrian | 35 | 5 | - | 1 | 1 | RGB | 640×360 | ✗ |
| ShanghaiTech [47] | 2017 | Surveillance | Pedestrian | 437 | 13 | - | 13 | 1 | RGB | 856×480 | ✗ |
| UCF-Crime [85] | 2018 | Online Surv. | Crime | 1900 | 12 | 1 | NA | NA | RGB | Multiple | ✓ |
| Street Scene [67] | 2020 | Surveillance | Traffic | 81 | 17 | - | 1 | 1 | RGB | 1280×720 | ✗ |
| IITB Corridor [75] | 2020 | Surveillance | Pedestrian | 358 | 10 | - | 1 | 1 | RGB | 1920×1080 | ✗ |
| XD-Violence [114] | 2020 | Films/Online | Violence | 4754 | 5 | 1 | NA | NA | RGB+Audio | 640×360 | ✓ |
| UBnormal [2] | 2022 | 3D modeling | Pedestrian | 543 | 20 | 2 | 29 | 8 | RGB | 1080×720 | ✓ |
| NWPU Campus [6] | 2023 | Surveillance | Pedestrian | 547 | 27 | 1 | 43 | 1 | RGB | Multiple | ✗ |
| CUVA [18] | 2024 | News/Online | Multiple | 1000 | 27 | 15 | NA | NA | RGB+Text | Multiple | ✓ |
| **MSAD (ours)** | 2024 | Online Surv. | Multiple | 720 | **35** | **20** | ∼**500** | **14** | RGB | Multiple | ✓ |

methods. A review on VAD benchmark datasets is presented in Appendix G. Table 1 presents a comparison of existing datasets from various perspectives.

## 2.1 Dataset deficiencies and biases

**From handcrafted to learned features.** Early VAD methods use traditional techniques such as background subtraction, optical flow, and handcrafted feature extraction [55, 92, 82, 77, 88], relying on appearance, motion, and texture to model motions and crowds [10, 122, 121, 5, 69]. However, these features are often insufficient due to the low resolution of benchmarks and limited training sources [3, 54, 93, 45], *e.g.*, Subway, UMN, UCSD Ped and CUHK Avenue, *etc*. Subsequent works [35, 41, 63, 111, 128] explore the combination of local/global features, spatial/temporal normalcy, *etc*. Nevertheless, these methods are ineffective when applied to different camera views and cannot adapt to unseen anomalies, when ShanghaiTech (13 camera views) is introduced. Consequently, emerging methods predominantly investigate the use of learned features, eliminating the need for handcrafted features and making them more adaptable to various camera viewpoints and new scenarios [46]. These methods mainly focus on creating new architectures or modules tailored to specific problems, and they can be broadly classified into four categories: reconstruction-based [4, 26, 33], prediction-based [43, 46, 24, 7, 73, 95], using classifiers [78, 58, 79, 80, 53, 33, 117], and scoring [37, 86, 32, 20, 48], ranging from simple architectures [99] to complex unified approaches [13].

**Challenges of learned features.** Most deep learning-based methods still carefully consider several aspects, such as visual appearance and motion, of human action [96, 43, 33, 105, 24, 100, 108, 98, 15, 97, 12]. While these works highlight the importance of appearance and motion in detecting mostly human-related anomalies [24, 31, 33], non-human-related anomalies remain under-explored due to the lack of solid benchmarks, even when the UCF-Crime dataset (1 non-human-related anomaly) is introduced. Recent works [4] have delved into creating end-to-end deep learning approaches and unified architectures rather than using separate modules or components in a traditional pipeline, as end-to-end solutions are easily accessible, usable, and deployable. However, deep learning solutions require a significant amount of training data, posing a significant concern, especially considering older and smaller datasets such as UCSD Ped [93] and CUHK Avenue [45]. Although efforts have been made in collecting large-scale datasets [85, 114] such as UCF-crime (video- and frame-level annotations for training and testing, respectively), an important issue lies in laborious video annotation, which is one of the main reasons why there haven't been as many large-scale VAD datasets published yet despite tons of data being publicly available on video sharing sites. A recent emerging few-shot learning framework [23] also encourages researchers to start looking at few-shot VAD models, due to (i) its fast adaptation to novel camera viewpoints/scenarios, (ii) relieving the training data hungry

issue, and (iii) its huge potential in real-world applications. Since then, even smaller datasets have proven to be helpful [46, 49].

**The beauty of few-shot learning.** Few-shot learning aims to adapt quickly to a new task with only a few training samples [106, 101, 102, 107]. One of the most widely recognized methods in VAD is the Few-shot Scene-adaptive Anomaly Detection (FSAD) model [46]. This model uses the meta-learning framework [23] to train a model using video data collected from various camera views within the same scenario, such as ShanghaiTech. The model is then fine-tuned on a different camera viewpoint within the university site (*e.g.*, UCSD Ped and CUHK Avenue). Although the trained model can be adapted to novel viewpoints, its adaptability is still confined to a specific scenario, such as the university street example. The absence of a multi-scenario dataset hampers the widespread application of the rapid adaptation capabilities of VAD models. Recent few-shot VAD methods include [46, 113, 49, 106, 107].

**Why self-supervised and weakly-supervised?** Traditional supervised learning methods require labeled data, which is often scarce or expensive to obtain for anomalies, even though normal video data are easy to obtain. This is where self-supervised and weakly supervised methods come into play. Anomaly samples are difficult to obtain, and it is challenging to define all types of anomalies (Table 1 shows earlier datasets have very limited anomaly types); therefore, state-of-the-art VAD methods are rarely trained using fully supervised approaches [72]. Self-supervised methods are proposed based on the assumption that any pattern that deviates from the learned normal patterns will be considered anomalies. Representative methods include reconstruction-based [26, 49, 61, 124], prediction-based [43, 46, 24, 7, 73, 95], and distance-based [30, 33, 68] solutions. While previous studies have emphasized the importance of memorizing normal patterns [4, 26, 33], numerous studies [85, 14, 129] have indicated that the assumption underlying the self-supervised paradigm is not always valid: (i) It is impractical to obtain all normal types from diverse scenarios with varied distributions (*e.g.*, crowded streets *vs*. empty parking lots). (ii) The boundary between normal and abnormal behavior is often ambiguous; even the same abnormal behavior may lead to different detection results under different scenarios. Therefore, weakly supervised VAD has emerged, training on both normal and abnormal samples using video-level annotations, *e.g.*, on UCF-Crime. This approach avoids the issue of frame-level or pixel-level annotations, which are time-consuming and labor-intensive; however, video-level annotations only indicate that the video contains anomalies, leaving the exact start and end of the anomaly unclear. Existing methods in this category often use pre-trained models, such as TSN [109], C3D [91], I3D [9], SwinTransformer [44], *etc.*, as an encoder to extract features.

**Expanding modalities for human-related anomaly detection.** Human-related anomaly detection solutions are not limited to the use of RGB videos. To efficiently extract motions, optical flow, precomputed on RGB videos, has been widely used as temporal or motion information for VAD [125]. Due to the heavy computational cost of optical flow and the redundant nature of RGB videos, researchers have started to explore alternative data modalities for efficient feature extraction [110]. Human pose estimation frameworks, like OpenPose [8], have made human skeleton sequences readily available from RGB videos. This not only introduces a new data modality but also addresses privacy concerns in human-related anomaly detection. The lightweight nature of skeletons [65, 36, 103] and informative spatio-temporal sequences have fostered many skeletal anomaly detection solutions [57, 31]. For example, [57] uses 2D human skeleton motions to detect anomalous human behavior in surveillance footage. Recently, LLMs and pretrained video caption models provide rich descriptions and prompts for video contents to assist VAD tasks [18, 50, 127]. Since then, multi-modal datasets have begun to emerge, with representative examples such as CUVA (RGB videos with text descriptions) in 2024.

**Advancements in multi-modal fusion.** A growing number of methods have begun integrating multi-modal information in recent years. Relying solely on optical flows [43, 104] or skeletons [30] may not accurately detect anomalies for two main reasons: (i) motions may not reflect relevant anomalies, and (ii) non-human-related anomalies may be treated as background. To address these challenges, several methods have emerged to comprehensively understand anomalies. These include the development of an audio-visual violence dataset and a new fusion method [114, 16], as well as the exploration of semantic information in VAD [13, 64, 116, 126]. One notable approach is a two-branch setup [116] (visual and language-visual branches) that utilizes a pre-trained visual-language model (*e.g.*, CLIP [66]). Additionally, the use of knowledge-based prompts as semantic information [64] and the fusion of text features extracted from a pre-trained video caption model with visual features [13]

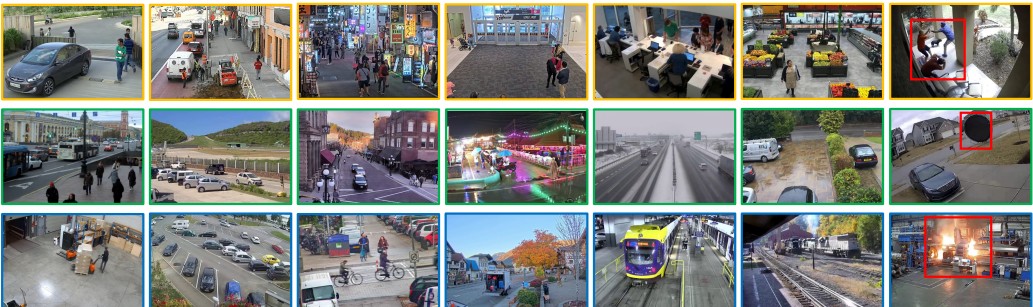

Figure 2: Our MSAD includes a diverse range of scenarios, both indoor and outdoor, featuring various objects, *e.g.*, pedestrians, cars, trains, *etc*. The first row shows different real-world common motions, while the second row demonstrates variations in weather and lighting conditions. The third row displays different moving objects. The last column shows human- and non-human-related anomalies.

have been explored. However, compared to single-modality methods, the improvement from these fusion approaches remains somewhat limited, indicating the need for further refinement [74, 13].

## 2.2 Discussion on model and dataset evolution

**Context-awareness.** Detecting individual objects or actions without considering context can introduce bias or errors, given that anomalies are defined based on their contextual relevance. For example, the NWPU dataset introduced in [6] is designed to detect scene-dependent anomalies and aims to predict these anomalies in advance, thereby enabling early warnings [21]. The robustness of VAD to various environmental variations such as lighting, weather, and road conditions becomes increasingly crucial for real-world applications. However, most popular benchmark datasets such as UCSD Ped, ShanghaiTech, and CUHK Avenue only simulate anomalies using humans and do not consider diverse and complex environmental conditions. This limitation makes it difficult for well-trained models to effectively detect real-world anomalies. Our dataset accounts for these environment variations, making it better aligned with real-world applications.

**Generalizability.** Existing algorithms often restrict themselves to detecting a handful of specific anomaly types. These models frequently suffer from poor generalizability, necessitating retraining for each specific target scenario or even a particular camera viewpoint, which subsequently increases computational cost. Additionally, many techniques are particularly vulnerable to various external factors such as illumination changes and complex backgrounds (*e.g.*, tree swaying and rainy), resulting in frequent false positives or negatives and consequently lowering the precision and reliability of VAD. Collecting and labeling anomalies remains challenging due to their rarity and diversity. While synthetic anomaly data can be generated using game engines across multiple scenarios with precise pixel-level annotations, high-quality real data is still required. Along with the datasets, establishing relevant benchmark evaluation metrics, such as more comprehensive evaluations, allows researchers to compare different methods and drive the development of better-performing algorithms. This motivates us to collect a new comprehensive benchmark dataset with thoughtfully designed evaluation protocols.

**Adaptability and reliability.** VAD faces the challenge of evolving definitions of anomalies over time, even for a specific camera viewpoint. For example, anomalies during the daytime and nighttime could differ, as could anomalies during workdays and weekends, necessitating adaptable and robust detection systems. These systems must continuously adapt to new video signals to maintain long-term reliability and accuracy, whether in single-scenario multiple viewpoints or multi-scenario settings. The robustness of VAD systems is significantly affected by the quality and size of the datasets used in training. High-quality, expansive datasets, combined with advanced algorithms, are essential to address the evolving nature of anomalies in VAD. Our experiments show the potential of our newly introduced dataset in tackling the challenges of detecting anomalies across diverse and dynamic surveillance scenarios. Notably, our dataset includes longer videos that capture the long-term evolution of video signals as well as anomalies.

**Interpretability and privacy concerns.** Below we list several limitations we observe during our review. First, only a few datasets in VAD contain multimodal data, which limits the development

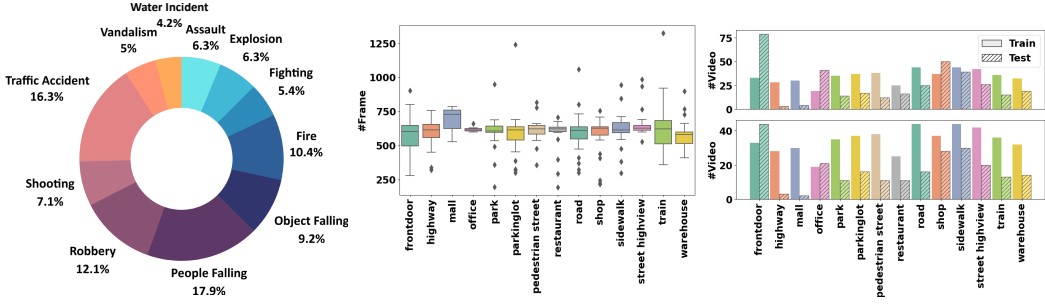

(a) The proportions of anomalies.  (b) Variations in frame numbers.  (c) Distributions of train/test splits.

Figure 3: The statistics of our MSAD dataset include: (a) a breakdown of main anomaly types and their respective percentages, (b) a boxplot illustrating frame number variations across scenarios in MSAD training set, and (c) the distributions of train/test splits across scenarios for two evaluation protocols (see Sec. 3 evaluation protocols): (*top* plot) generalizability and adaptability, and (*bottom* plot) practical applicability and effectiveness.

of relevant multimodal methods. XD-Violence introduced audio information and demonstrated the positive impact of audio-visual fusion. Therefore, exploring multimodal datasets and better fusion strategies that maximize the benefits of each modality could be a promising research direction. We plan to extend our dataset to include more modalities, such as audio and video descriptions. Second, most VAD models adopt a data-driven, end-to-end pipeline. Although effective, the learned features are often not interpretable, which may hinder deployment in real-world applications due to security and safety concerns. Developing explainable and interpretable models provides insights into the underlying reasoning for detected anomalies, making the automated system easier for end users to accept. Lastly, it is crucial to address privacy and ethical concerns associated with VAD, particularly regarding data collection in public spaces for surveillance. Balancing VAD performance while removing personally identifiable information, such as faces, poses, or gaits of individuals, and license plates of vehicles, would be an interesting area of exploration. We leave these for future work. More discussions are provided in Appendix E.

## 3 A Multi-Scenario Dataset

Unlike many datasets with fixed camera viewpoints and limited scenarios, our Multi-Scenario Anomaly Detection (MSAD) dataset boasts a broader range of scenarios and camera viewpoints (refer to Table 1 for a comparison). The process for collecting our dataset is detailed in Appendix A.

**Viewpoints *vs*. scenarios.** Traditional datasets commonly define a scene as the perspective captured by a camera, often referred to as a camera viewpoint. For instance, ShanghaiTech [47] comprises 13 scenes, representing videos recorded from 13 distinct camera viewpoints at a university. However, relying solely on the 'scene' or 'camera view' concept is insufficient for robust anomaly detection. Models trained on multi-view videos per scenario face limitations and struggle to adapt to new scenarios, particularly when confronted with novel camera viewpoints. To overcome this limitation and enhance multi-scenario anomaly detection, we introduce the 'scenario' concept to describe different environments. Our dataset encompasses 14 distinct scenarios, including front door areas, highways, malls, offices, parks, parking lots, pedestrian streets, restaurants, roads, shops, sidewalks, overhead street views, trains, and warehouses. Fig. 2 showcases some video frames from these diverse scenarios. As depicted in the figure, our dataset features more realistic scenarios compared to existing benchmarks. It covers a broad spectrum of objects and motions, along with multiple variations in the environment, such as changes in lighting, diverse weather conditions, and more.

**Human- *vs*. non-human-related anomalies.** Previous studies have primarily characterized anomalies as human-related behaviors, encompassing activities like running, fighting, and throwing objects. This emphasis on human-related anomalies stems from their greater prevalence in real-world scenarios. However, enumerating all types of anomalies in diverse real-world contexts poses a significant challenge. To address this, our dataset is further categorized into two principal subsets: (i) Human-related: This subset features scenarios where only human subjects engage in activities, facilitating

human-related anomaly detection. For instance, scenarios involve people interacting with various objects like balls or engaging with vehicles such as cars and trains. (ii) Non-human-related: This subset includes scenarios related to industrial automation or smart manufacturing, denoting the use of control systems for operating equipment in factories and industrial settings with minimal human intervention. This subset is designed for detecting anomalies, *e.g.*, water leaks, fires, *etc*.

**Dataset statistics.** Figure 3 provides statistics for our MSAD dataset. Our dataset features a wide range of anomalies, including 35 human-related anomalies such as people falling down, school fights, street fights, facility vandalism, and shop robberies, as well as 20 non-human-related anomalies like water leaks, floods, factory fires, trees falling down, and office fires. Figure 3a illustrates the proportions of main anomalies in the dataset. Note that we group anomalies such as street fights and school fights into the main category "Fighting" for better visualization purposes. We provide all detailed anomaly types in Appendix B. Our dataset contains 720 videos (447,236 frames in total), with an average video length of 621.16 frames, and some videos extending up to 6,026 frames. The frames are extracted from the original videos at a rate of 30 FPS.

**Evaluation protocols.** To ensure a fair comparison between different algorithms, we provide frame-level annotations during testing stage. More detailed information can be found in Appendix B. We design two evaluation protocols for using our MSAD dataset:

i. Train on 360 normal videos from 14 scenarios and test on the remaining 120 normal videos and 240 abnormal videos. Figure 3b shows a boxplot of frame number variations across scenarios in training set. This protocol is suitable for evaluating self-supervised methods.

ii. Train on 360 normal and 120 abnormal videos, and test on 120 normal and 120 abnormal videos. During training, we only provide video-level annotations. This protocol is suitable for evaluating weakly-supervised methods trained with our video-level annotations.

The design of these two evaluation protocols aims to facilitate a comprehensive evaluation by considering different models and training schemes. Such a design has not been considered before. Our dataset usage and maintenance are detailed in Appendix C. Figure 3c (*top* plot) shows the train/test split distributions for Protocol i, while (*bottom* plot) shows the distributions for Protocol ii.

We also apply Protocol i to both Few-shot Scene-adaptive Anomaly Detection (FSAD)[46] and our Scenario Adaptive Anomaly Detection (SA$^2$D). A detailed description of our SA$^2$D is provided in Appendix D. Our method focuses on evaluating how MSAD can contribute to fast adaptation not only to new camera viewpoints but also to new scenarios in VAD.

## 4 Experiment

In this section, we design two main sets of experiments to explore the capabilities of our MSAD dataset. For generalizability and adaptability evaluation, we use few-shot methods to conduct both cross-view (single-scenario) and cross-scenario evaluations under Protocol i, which helps assess the model's ability to adapt to new viewpoints and scenarios with limited training data. For practical considerations and popularity, we select some representative weakly supervised methods for evaluation using Protocol ii. This protocol evaluates the practical applicability and effectiveness of these methods using our dataset, reflecting common practices in the field. All experiments are conducted using the National Computational Infrastructure (NCI) Gadi, with one V100 GPU allocated for each experiment. Additional evaluations can be found in Appendix F.

### 4.1 Generalizability and adaptability

**Setup.** We first show the superior performance of our scenario-adaptive model by comparing SA$^2$D with the FSAD model [46] on CUHK Avenue. To demonstrate the enhanced anomaly detection performance of our MSAD dataset, we train two adaptive models: one using ShanghaiTech, and the other leveraging our MSAD. For all experiments, we maintain $N = 7$ and $K = 10$ to ensure a fair comparison. Our training process spans over 1500 epochs. Throughout all evaluations, we consider both Micro and Macro AUC scores. We conduct two sets of experiments: (1) In the *single-scenario / cross-view evaluation*, the model is trained and tested on the same scenario. To facilitate a comparison with the view-adaptive anomaly detection model [46], we partition the ShanghaiTech dataset into 7 scenes (views), specifically, scene 2, 4, 7, 9, 10, 11, and 12 for training as in [46]. Subsequently,

Table 2: Experimental results on single-scenario evaluation. On ShanghaiTech (ShT), only 7 views are used during training and the rest views are individually used for testing. The notation ShT-$v*$ denotes the use of different camera views.

| Test view | Train | FSAD [46] | | Train | FSAD [46] | | SA$^2$D (ours) | |
|---|---|---|---|---|---|---|---|---|
| | | Micro | Macro | | Micro | Macro | Micro | Macro |
| ShT-$v1$ | ShT (7 views) | 61.36 | 55.34 | MSAD | 63.74 | 62.92 | **68.96** | **77.89** |
| ShT-$v3$ | | 26.51 | 26.58 | | 64.39 | 62.56 | **67.59** | **73.43** |
| ShT-$v5$ | | 53.40 | 53.32 | | 55.04 | 54.63 | **55.74** | **54.02** |
| ShT-$v6$ | | 78.36 | 78.27 | | 70.26 | 71.02 | **75.47** | **72.35** |
| ShT-$v8$ | | 50.02 | 52.54 | | 59.97 | 57.45 | **60.85** | **61.52** |

Table 3: Cross-scenario evaluations using FSAD [46] and SA$^2$D (ours) trained on ShT and MSAD.

| Train | Test | AUC | |
|---|---|---|---|
| | | Micro | Macro |
| ShT | Ped2 | 57.38 | 58.36 |
| | CUHK | 69.98 | 78.32 |
| | MSAD | **63.92** | **64.92** |
| MSAD | Ped2 | **70.35** | **65.74** |
| | CUHK | **79.57** | **84.49** |
| | MSAD | **69.96** | **69.60** |

the remaining 5 camera views are individually used for testing purposes.[4] (2) In the *cross-scenario evaluation*, the model is trained on one scenario and tested on a completely different one. For model training, we use either ShanghaiTech or our MSAD.

**Single-scenario evaluation.** As illustrated in Table 2, our model is trained on seven camera views of the ShanghaiTech dataset and tested on the remaining views, such as $v1$, $v3$, and so forth. The performance in cross-view evaluation falls short compared to our SA$^2$D. Notably, our SA$^2$D, trained on our MSAD and tested on ShanghaiTech, exhibits significantly superior performance compared to the view-adaptive model. This disparity in performance may stem from: (i) the constraint of limited training views, preventing the model from effectively adapting to novel viewpoints, (ii) the camera view being tested on at ShanghaiTech is significantly different, almost equivalent to a novel scenario concept. Therefore, the view-adaptive model is unable to adapt to such novel scenarios.

**Cross-scenario evaluation.** Our model, trained on MSAD, demonstrates superior performance compared to the view-adaptive model (see Table 3, Ped2 and CUHK refer to UCSD Ped2 and CUHK Avenue datasets). For instance, on CUHK Avenue, our model outperforms the view-adaptive model by 9.6% and 6.2% for Micro and Macro evaluation metrics, respectively. Furthermore, our SA$^2$D, trained and tested on MSAD, consistently achieves excellent performance. It's important to note that our MSAD test set is distinct from the MSAD training set, covering a diverse range of novel scenarios. These results underscore the robustness of our model in cross-scenario evaluations. However, we also observe a performance drop on ShanghaiTech ($v6$). The decline in performance is attributed to the dataset deviating from real-life scenarios, as it categorizes biking and driving as anomalies, a classification that does not align with reality.

**Insights on single- and cross-scenario evaluations.** Based on the aforementioned experiments, we can deduce that a model trained on intricate real-world scenarios exhibits superior generalization. This stems from the fact that real-world models are frequently influenced by the surrounding environment, encompassing elements like fluctuating traffic patterns, dynamic electronic displays, and the movement of trees in the wind. The model must discern the nuances of anomaly detection within a dynamic environment and comprehend the dynamics of objects and/or performing subjects within it. Our MSAD dataset provides a comprehensive representation of real-world scenarios.

**Scene information is implicitly used as weak supervision.** The scene information provided is used in the few-shot sampling strategy, where each scenario is treated as a group for sampling purposes. This allows the model to learn information from multiple scenarios in a balanced manner (see Fig. 6 in Appendix D). In our proposed model, SA$^2$D, scene information is used during the sampling process to form $N$-way $K$-shot learning. Table 2's last four columns compare the performance (i) without scene information (FSAD) and (ii) with scene information (our SA$^2$D) on our MSAD dataset (note that both models are trained using MSAD). The primary difference between FSAD and SA$^2$D is that SA$^2$D uses scene information during sampling to enhance the $N$-way $K$-shot learning framework. As shown in Table 2, our sampling strategy (Appendix D) significantly boosts SA$^2$D's performance across all five test splits on different camera viewpoints in ShanghaiTech.

---

[4]There are no test videos for scene 13, so we only consider 5 views during testing.

Table 4: Comparison of six methods with varying backbones on UCF-Crime, ShanghaiTech, and our MSAD dataset using three popular backbones: C3D, I3D, and SwinTransformer (SwinT).

| | Venue | UCF-Crime | | | ShanghaiTech | | | MSAD | |
|---|---|---|---|---|---|---|---|---|---|
| | | C3D | I3D | SwinT | C3D | I3D | SwinT | I3D | SwinT |
| MIST [22] | CVPR 2021 | 81.40 | 82.30 | - | 93.13 | 94.83 | - | - | - |
| RTFM [90] | ICCV 2021 | 83.28 | 83.14 | 83.31 | 91.51 | 97.94 | 96.76 | 86.65 | 85.67 |
| MSL [40] | AAAI 2022 | 82.85 | 85.30 | 85.62 | 94.23 | 95.45 | 97.32 | - | - |
| UR-DMU [129] | AAAI 2023 | 82.65 | 86.19 | 83.74 | 94.67 | 96.15 | 95.71 | 85.02 | 72.36 |
| MGFN [14] | AAAI 2023 | 82.37 | 83.44 | 84.30 | 90.82 | 93.97 | 93.58 | 84.96 | 78.94 |
| TEVAD [13] | CVPRW 2023 | 83.39 | 84.54 | 84.65 | 92.05 | 98.10 | 97.63 | 86.82 | 83.60 |

## 4.2 Practical applicability and effectiveness

**Setup.** We focus on weakly-supervised learning methods from a practical perspective. We select six recent representative methods for evaluation: MIST [22], RTFM [90], MSL [40], UR-DMU [129], MGFN [14], and TEVAD [13]. These methods provide open-source implementations, allowing us to explore the impact of different backbone choices. Each method has its unique focus. MIST introduces a multiple instance self-training framework to refine task-specific discriminative representations using only video-level annotations. RTFM trains a feature magnitude learning function to identify positive instances, enhancing the robustness of the multiple instance learning approach. MSL uses multi-sequence learning to predict both video-level anomaly probabilities and snippet-level anomaly scores. UR-DMU focuses on learning representations of normal data while extracting discriminative features from abnormal data for a better understanding of normal states. MGFN proposes a glance-and-focus network to amplify the discriminative power of feature magnitudes across different scenes. TEVAD uses both visual and text features to complement spatio-temporal features with semantic meanings of abnormal events. These methods enable us to conduct a fair and comprehensive evaluation across various aspects.

In line with the standard practice in [90, 40, 129], we use C3D [91], I3D [9], SwinTransformer [44] pretrained on Kinetics-400, as backbone networks for feature extraction. Although the same backbone is used, different methods may extract features of varying dimensions. For example, UR-DMU and MSL extracts 1024-dim features, whereas other methods extract 2048-dim features when using I3D as the backbone. To ensure a fair comparison, following RTFM, we extract 4096-dim features from the 'fc6' layer of C3D, 2048-dim, 10-crop features from the 'mix 5c' layer of I3D, respectively. Additionally, following MSL, we extract 1024-dim features from SwinTransformer. We apply these feature extraction processes on ShanghaiTech, UCF-Crime and our MSAD. Then we use these extracted features as inputs of different methods. We rigorously maintain original parameters to reproduce reported results from previous studies, and conduct our own evaluations using Protocol ii. We use the Micro AUC metric for evaluation.

**Evaluation on methods.** Table 4 shows the results. As shown in the table, there is no single best performer across all three datasets. However, on ShanghaiTech with I3D as the backbone, TEVAD emerges as the top performer. This demonstrates the efficacy of using text descriptions in video anomaly detection. The second-best performer is RTFM, followed by UR-DMU.

**Evaluation on backbones.** For MSL, we observe that SwinTransformer outperforms I3D, and I3D outperforms C3D. Specifically, on UCF-Crime, SwinTransformer as a backbone performs better than using I3D. However, on our MSAD dataset, I3D performs better than SwinTransformer. The potential reason for this behavior is that UCF-Crime generally contains longer videos, and SwinTransformer is good at capturing long-term motions.

**MSAD is sensitive to the choice of backbones.** We notice that our dataset is sensitive to the choice of backbones, particularly for methods like UR-DMU and MGFN (Table 4). Existing backbones, such as C3D, I3D, and SwinTransformer, have demonstrated strong performance in video anomaly detection on current datasets (see Table 4 for results on UCF-Crime and ShanghaiTech). However, these datasets have significant limitations: (i) they cover a very limited range of scenarios, motion dynamics, and anomaly types, and (ii) most anomaly detection methods rely on action recognition pre-trained models as backbones for feature extraction, particularly for human-related anomalies. As a result, these backbones tend to perform well because the anomaly detection datasets predominantly feature human-related motions. In contrast, our dataset introduces more challenges in terms of scenarios,

camera viewpoints, and motion dynamics, including both human and non-human-related motions. Overall, using I3D as the backbone yields better results compared to using SwinTransformer. RTFM is robust to the choice of backbones, with the performance gap between I3D and SwinTransformer being within 1%.

Our dataset can be used for selecting appropriate model backbones or exploring more powerful backbone networks that do not overly depend on existing action recognition models. Additionally, our dataset advances video anomaly detection by considering a wider range of scenarios and a broader spectrum of anomaly types.

### 4.3 Limitations of current methods on our new dataset

Here, we discuss the limitations of current methods and provide additional insights on our MSAD dataset.

**I3D *vs*. SwinTransformer.** Using I3D as a backbone generally outperforms SwinTransformer. Although SwinTransformer excels in capturing local and global spatial features, its reliance on attention mechanisms for temporal modeling may not be as finely tuned to detect subtle anomalies. I3D, designed to maintain temporal consistency across frames, is better suited for detecting anomalies that are temporally localized, *e.g.*, sudden object appearances or unexpected behaviors. SwinTransformer's attention-based approach may lack the temporal coherence needed for such tasks. Additionally, I3D's focus on motion and spatiotemporal features aligns well with anomaly detection requirements, especially in motion-based anomalies and subtle temporal variations present in our dataset.

**Boosting existing methods.** Models trained on our dataset exhibit better generalizability and adaptability, particularly under few-shot settings (Table 2), showing superior performance on unseen camera viewpoints. Our SA$^2$D model, trained on MSAD (Table 3), significantly boosts performance. Additionally, our dataset enhances the performance of existing anomaly detection methods (Table 7 in the Appendix). Models trained on MSAD consistently achieve the best performance on anomalies such as fighting, fire, object falling, shooting, traffic accidents, and water incidents, addressing the challenge of lacking a comprehensive benchmark dataset for model training.

**No single best performer on MSAD.** RTFM and TEVAD are more robust to backbone changes than UR-DMU and MGFN (see Table 4). This suggests that existing methods may lack comprehensive evaluations across diverse scenarios. Our dataset provides a solid benchmark for training, testing, and evaluating superior models.

**Supporting multi-scenario anomaly detection.** Our dataset benefits multi-scenario anomaly detection, enabling systematic evaluation across multiple scenarios (Table 8 in the Appendix). Recent methods trained on MSAD generally achieve better performance across all 14 scenarios, demonstrating that existing works have primarily focused on single-scenario detection. Our dataset offers a robust foundation for training superior models for multi-scenario anomaly evaluation.

**Rethinking anomaly detection.** Our dataset encourages rethinking the correctness and trustworthiness of anomaly detection methods. Cross-dataset evaluation (Table 9 in the Appendix) reveals that existing datasets (Table 1) may suffer from unrealistic anomaly definitions. For instance, models pre-trained on MSAD perform poorly on ShanghaiTech but better on CUHK and UCSD Ped2, highlighting the misalignment of anomaly categories like biking and driving with reality.

## 5  Conclusion

In this paper, we provide a concise review and identify several practical issues in video anomaly detection, particularly the lack of comprehensive datasets with diverse scenarios. To address these challenges, we introduce the Multi-Scenario Anomaly Detection (MSAD) dataset. This dataset includes diverse motion patterns and challenging variations, providing a robust foundation for developing superior models. Alongside the dataset, we present our SA$^2$D model, which uses a few-shot learning framework to efficiently adapt to new concepts and scenarios. Experimental results demonstrate the model's robustness, excelling not only in new camera views of the same scenario but also in novel scenarios. Our contributions offer valuable resources and insights to advance the field of video anomaly detection, addressing current challenges and setting the stage for future research directions.

## Acknowledgments and Disclosure of Funding

Liyun Zhu conducted this research under the supervision of Lei Wang for his final year master's research project at ANU. Liyun Zhu and Arjun Raj are recipients of The Active Intelligence Research Challenge Award. This work was also supported by the NCI Adapter Scheme Q4 2023 and the NCI National AI Flagship Merit Allocation Scheme, with computational resources provided by NCI Australia, an NCRIS-enabled capability supported by the Australian Government.

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

# A  Dataset collection process

To facilitate the data collection, we use a script to automatically download relevant videos from YouTube, Itemfix, and Bilibili and trim them into short clips. We ensure that all the videos are selected and used in accordance with the fair dealing provisions of the Australian Copyright Act 1968. Our approach is designed to adhere strictly to these legal frameworks, ensuring that the use of copyrighted material is only limited to research purposes. This careful consideration helps to protect the rights of the original content creators while allowing us to utilize the material in a legally compliant manner.

Real-world scenarios present greater challenges due to diverse weather and lighting conditions, as well as unpredictable objects that can impact detection, such as dynamic electronic advertising screens or moving car headlights during nighttime in video surveillance. Our dataset considers these variations. Our test set covers a broad spectrum of abnormal events observed in diverse scenarios, including both human-related and non-human-related anomalies, such as fire incidents, fighting, shooting, traffic accidents, falls, and other anomalies. Our test set exclusively contains video frame-level annotations. We apply a rigorous filtering process to ensure the quality and appropriateness of the collected videos. The following criteria are used to remove unsuitable videos:

i. Quality considerations:

    (a) Low resolution: videos that do not meet our resolution standards are excluded.

    (b) Grayscale videos: only color videos are retained to ensure richness in visual information.

    (c) Moving camera views: videos with unstable or moving camera perspectives are excluded to maintain consistency in viewpoint, as we focus on static camera video anomaly detection (VAD).

ii. Content considerations:

    (a) Excessive text overlays: videos with many text overlays that obscure the visual content are removed.

    (b) Potentially invasive content: videos that potentially invade privacy, such as those showing identifiable faces, are excluded.

    (c) Violent content: videos that are overly violent, depict violence against children, or contain graphic content are removed to ensure ethical standards.

    (d) Political content: videos containing political content are excluded to avoid any potential biases or controversies.

This thorough process ensures that our dataset is of high quality, ethically sound, and suitable for research purposes in VAD.

While we acknowledge that surveillance videos can raise privacy and violence concerns, the potential risks are magnified when anomalies occur. A real-world anomaly detection dataset serves as a necessary complement, despite the inherent drawbacks of surveillance. However, it is crucial to implement appropriate regulations to minimize the potential negative impact of datasets. Our MSAD dataset is intended exclusively for academic research. Researchers who wish to access the dataset must complete the online form and agree to the terms and conditions.

**Identifiable information.** To eliminate any identifiable information, we consider the following options. (i) Face blurring: Since in surveillance video footage, the facial regions of individuals are often small and blurry, or the camera angles typically capture side or rear views, it is challenging to extract identifiable information about individuals. To eliminate any identifiable information that might be visible, we apply an automatic blurring script[5] to anonymize all the faces and car licenses in the videos of the MSAD dataset, then we check the whole dataset manually to ensure that all the identity information is almost eliminated. The blurred videos have limited impact to the detection performance while reducing the invasion of privacy. We provide the blurred version of our MSAD dataset. (ii) Extracted features: We have also released the extracted video features with different backbones, *e.g.*, I3D, SwinTransformer, *etc*., to the research community to promote privacy preservation. We have noticed that some weakly-supervised methods in recent years use extracted video features as inputs [39, 90, 14, 13, 116]. Training and evaluation with extracted video features is lightweight while protecting privacy. (iii) Deepfake technique: We also explore using deepfake techniques to

---

[5] https://github.com/ORB-HD/deface

remove identifiable features from the original videos, aiming to preserve characteristics such as age and gender while minimizing information loss compared to face blurring. However, we encounter challenges due to the diverse resolutions and camera angles, which make deepfake difficult. We plan to further investigate this approach in future work.

**Bias.** To ensure diversity in our dataset, we have extensively collected videos from various regions and countries. The purpose of collecting videos from diverse regions and countries is to advance anomaly detection technologies globally, benefiting communities worldwide. We conduct searches on platforms such as YouTube using text queries that combine various anomaly types (*e.g.*, fighting, fire, people falling, *etc*.) with diverse region and country-specific terms. To maximize the diversity of videos retrieved from around the world, we also utilize queries in multiple languages using translators. The collected videos represent a broad spectrum of ages, genders, ethnicities, and scenarios, which helps to reduce potential biases. By incorporating such a diverse array of content, our goal is to create a balanced and representative dataset that accurately mirrors the complexity and diversity of real-world situations.

# B    Further dataset details

**Anomaly types.** Table 5 shows the details of the main anomalies, detailed anomalies per main anomaly type, and their relevant domains. As shown in the table, our dataset encompasses a broad range of both human-related and non-human-related anomalies, spanning multiple domains. A boxplot showing frame number variations across scenarios in the whole MSAD dataset is provided in Fig. 4a.

**Resolution.** All the videos we've collected are high resolution, such as $1920 \times 1080$ (see Fig. 4b). Each video is captured from a fixed camera view with a standard frame rate of 30. Our dataset offers several advantages: (i) Most existing cameras in industry surveillance feature high resolution, thanks to advanced camera technologies. (ii) High-resolution videos provide more detailed information for anomaly detection tasks, particularly when the performing subject is distant from the camera, or the object being interacted with is relatively small. (iii) Existing anomaly detection models trained on low-resolution videos often suffer from poor detection performance, especially in the presence of tiny or fine-grained motion patterns. Therefore, our dataset is better suited for training a more effective anomaly detection model.

**Video/anomaly duration.** As shown in figure 4c, in our dataset, the majority of all videos (blue) have durations between 10 and 30 seconds, with a peak around 20 seconds. There is a sharp decline in the count of videos longer than 30 seconds. While the overall distribution (blue) and anomaly videos (orange) follow a similar trend, precise anomalies (green) show distinctive peaks at shorter durations, suggesting that anomalies are very rare and brief. This indicates that our dataset effectively captures essential and engaging moments within concise time frames, making it highly valuable for quick insights and analyses. The distribution highlights the efficiency and focus of the captured events.

**Scenario and anomaly details.** Fig. 4d shows more detailed distributions per anomaly type per scenario. It should be noted that not all anomaly types have bar lines because there are no scenarios associated with them. The plot shows a varying set of anomaly video percentages. In our dataset, we have endeavored to give a comprehensive collection of videos with varying anomaly lengths in them.

**Motion visualisation.** In figure 5, we present visualizations of some anomaly frames from our MSAD dataset, along with their corresponding mean motion maps. These maps are generated by averaging the frame difference maps from consecutive frames. The intensity and distribution of motion vary across different types of anomalies. For instance, continuous anomalies like fighting and robbery exhibit a widespread motion pattern, while abrupt incidents such as robbery and object falling display more concentrated motion. The maps also reveal specific directional movements, such as the downward motion evident in the object falling and people falling anomalies. Our dataset is designed to include a diverse array of videos, capturing a wide range of these attributes.

**Evaluation metrics.** Existing anomaly detection datasets are annotated in different ways, including pixel-level, frame-level and video-level annotations. Different annotation methods correspond to different evaluation metrics. For example, frame-level annotation often uses area under the curve (AUC) as the evaluation metric. As discussed in [25], there are two kinds of AUC score, namely

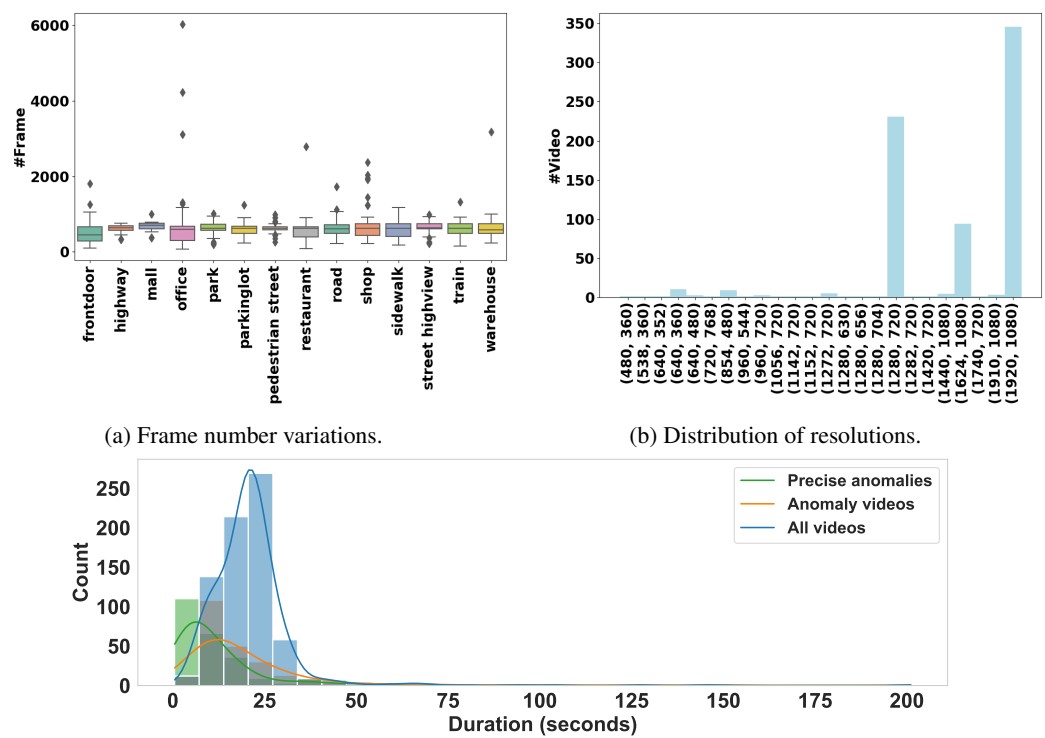

(a) Frame number variations.

(b) Distribution of resolutions.

(c) Distribution of video durations. The histogram displays video durations for actual anomaly video length (precise anomalies in green), anomaly videos (orange), and all videos (blue).

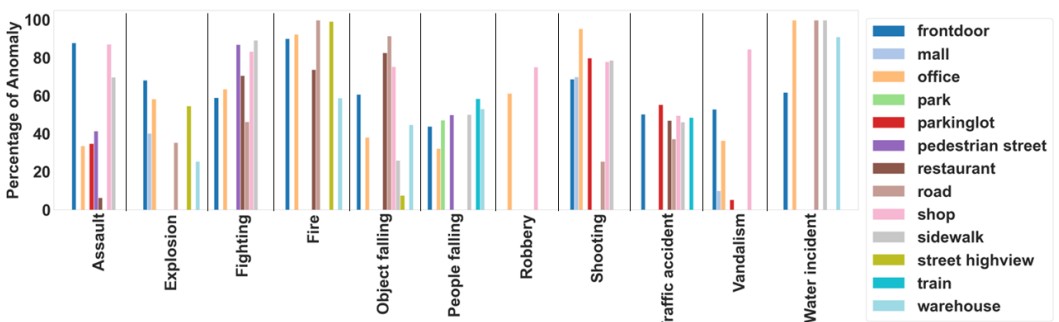

(d) The bar chart illustrates the percentage of anomalies in the anomaly videos grouped by anomaly types and scenarios. The black lines separate different types of anomalies.

Figure 4: Distributions of (a) frame number variations, (b) video resolutions, (c) video durations, and (d) detailed distributions per anomaly type per scenario in the entire MSAD dataset are presented. Our main paper presents the distribution of frame number variations in the training set (see Fig. 3b).

Table 5: Details of our MSAD dataset. Our dataset has a broad range of both human-related and non-human-related anomalies, spanning multiple domains.

| Main Anomaly Types | Detailed Anomaly Types | Domain |
|---|---|---|
| | | |

**Human-related anomaly**

| | Main Anomaly Types | Detailed Anomaly Types | Domain |
|---|---|---|---|
| | Assault | Assault on street | Crime |
| | | Assault in office | Crime |
| | | Other assault | Crime |
| | Fighting | Fighting on street | Violence |
| | | Fighting in a restaurant | Violence |
| | | Fighting in a shop | Violence |
| | | Fighting in front of a door | Violence |
| | | Fighting indoors | Violence |
| | | Other fighting | Violence |
| | People Falling | People falling to ground | Pedestrian |
| | | People falling into pool | Pedestrian |
| | | People falling from high places | Pedestrian |
| | | People falling into subway | Pedestrian |
| | | Other people falling | Pedestrian |
| Human-related anomaly | Robbery | Shop robbery | Crime |
| | | Office robbery | Crime |
| | | Theft | Crime |
| | | Car theft | Crime |
| | | Other robbery | Crime |
| | Shooting | Shooting on the road | Crime |
| | | Shooting indoors | Crime |
| | | Holding a gun | Crime |
| | | Other shooting | Crime |
| | Traffic Accident | Car falling | Traffic |
| | | Car crash | Traffic |
| | | Speeding | Traffic |
| | | Car rushing into building | Traffic |
| | | Car crash with people | Traffic |
| | | Car crash with object | Traffic |
| | | Car crash with train | Traffic |
| | | Motorcycle crash | Traffic |
| | | Other traffic accident | Traffic |
| | Vandalism | Vandalizing glass | Violence |
| | | Vandalizing door | Violence |
| | | Other vandalism | Violence |
| Non-human-related anomaly | Explosion | Street explosion | Emergency |
| | | Firework explosion | Emergency |
| | | Factory explosion | Emergency |
| | | Indoor explosion | Emergency |
| | | Other explosion | Emergency |
| | Fire | Smoke | Emergency |
| | | Factory fire | Emergency |
| | | Building on fire | Emergency |
| | | Bush fire | Emergency |
| | | Other fire | Emergency |
| | Object Falling | Strong wind | Natural hazard |
| | | Object falling in home | Emergency |
| | | Tree falling | Emergency |
| | | Large objects falling | Emergency |
| | | Glass falling | Emergency |
| | | Other objects falling | Emergency |
| | Water Incident | Flood | Natural hazard |
| | | Water leakage | Emergency |
| | | Heavy rain | Natural hazard |
| | | Other water incidents | Emergency |

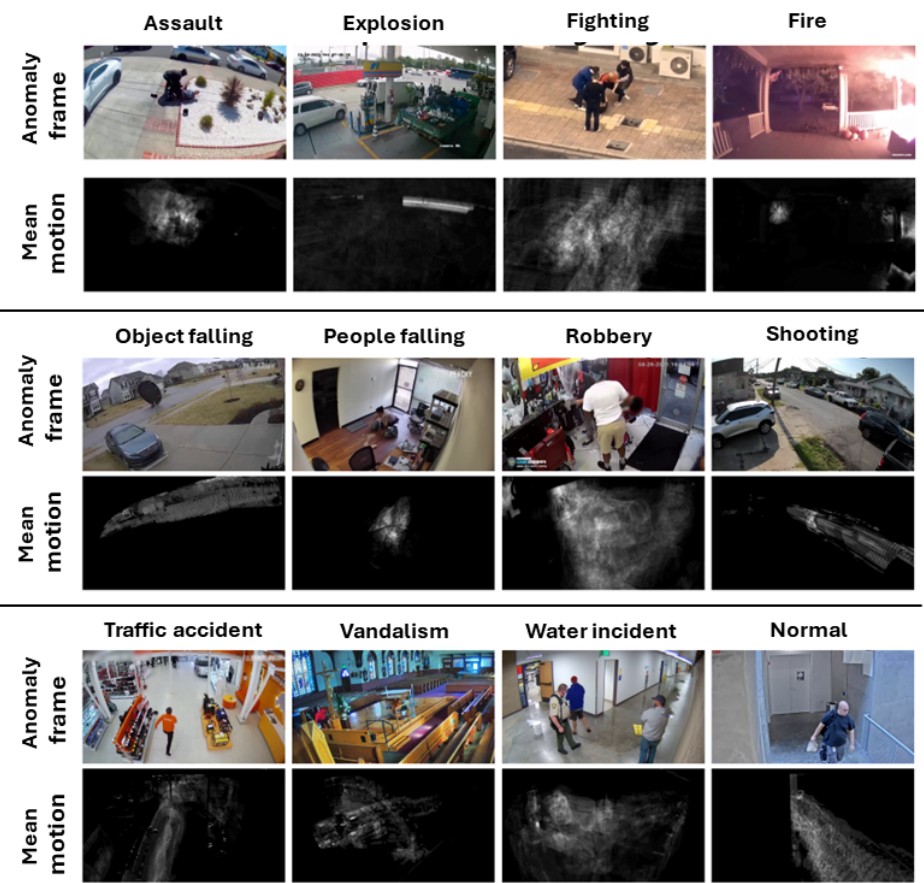

Figure 5: Visualizations of anomaly frames and their corresponding mean motion maps. The mean motion map is calculated by averaging the frame difference maps obtained from pairs of consecutive frames.

Micro-AUC and Macro-AUC. The former concatenates the frames from all the videos and computes the overall AUC value, and the latter computes the AUC value for each video and averages it.

For pixel-level annotations, Ramachandra *et al.* [67] introduced two evaluation metrics: the Region-Based Detection Criterion (RBDC) and the Track-Based Detection Criterion (TBDC). These metrics are designed to prioritize the false positive rate on both temporal and spatial dimensions. This consideration stems from the fact that anomalies in videos often extend across multiple frames. Hence, detecting anomalies in any segment and reducing the false detection rate holds significance for VAD systems.

To ensure a fair comparison between different algorithms, we use frame-level annotations for test. To be more precise, when the frame contains abnormal regions, we consider the frame as anomaly to obtain frame-level label. It is worth noting that many methods only use frame-level AUC as the evaluation metric. However, besides detecting anomalous events, it is crucial to avoid misclassifying normal events as anomalies as much as possible in practical applications. Therefore, in order to accurately evaluate the model's performance, we consider both frame-level AUC and false positive rate.

## C   Dataset usage and maintenance

VAD datasets serve multiple purposes within the research community. Primarily, they are used for training and evaluating machine learning models. By providing labeled examples of normal and anomalous events, these datasets enable the development of effective anomaly detection algorithms.

Researchers also use these datasets to benchmark the performance of different algorithms, facilitating consistent and fair comparisons across various methods. Additionally, the inclusion of diverse scenarios allows for cross-scenario testing, assessing the generalizability of anomaly detection models across different environmental conditions. Furthermore, these datasets are valuable for feature extraction and analysis, providing insights into the nature of anomalies and enhancing the interpretability of detection models.

The use of our dataset is governed by the following terms and conditions:

  i. Without the expressed permission from our research group members, any of the following will be considered illegal: redistribution, derivation or generation of a new dataset from this dataset, and commercial usage of any of these videos in any way or form, either partially or in its entirety.

  ii. For the sake of privacy, images of all subjects in any of these videos are only allowed for demonstration in academic publications and presentations.

This dataset is released for academic research only and is free to researchers from educational or research institutes for non-commercial purposes.

Ensuring the reliability and relevance of VAD datasets requires ongoing maintenance efforts. We will expand our dataset by adding more scenarios and anomaly types to cover more domains. We will continue collecting high-quality videos to advance static camera VAD. The limitation of our work is presented in Sec. H.

To address concerns about deprecated videos, we will regularly monitor the sources of our dataset to ensure that the videos or data links remain valid. This process may involve automated scripts that periodically verify online links and alert us if any sources become inaccessible. If the original video source becomes unavailable, we will seek alternative sources or mirrors that host the same content. Maintaining a list of alternative sources will help preserve the dataset's integrity. However, if no alternative sources are found, we may consider removing the unavailable content from our dataset to ensure that it remains up-to-date and reliable.

Additionally, we will update the experimental results on our dataset website if any videos become unavailable. We will also document the missing videos for transparency and fairness in future research comparisons. We will provide a detailed datasheet to facilitate a more comprehensive assessment of the benchmark and baselines. We have provided our script for downloading the videos, as well as extracted features from backbone networks such as I3D and SwinTransformer. We will also release our full evaluation framework to allow interested researchers to further explore our multi-scenario datasets in the project website. We believe that our efforts provide a solid foundation for future research on real-world anomaly detection, ultimately benefiting our community.

## D   Scenario-adaptive anomaly detection framework

Below we present our Scenario-Adaptive Anomaly Detection (SA$^2$D) model. First, we introduce the notations.

**Notations.** $\mathcal{I}_K$ represents the index set $1, 2, \cdots, K$. Calligraphic mathcal fonts denote tensors (*e.g.*, $\mathcal{V}$), capitalized bold symbols are matrices (*e.g.*, $\boldsymbol{F}$), lowercase bold symbols denote vectors (*e.g.*, $\boldsymbol{x}$), and regular fonts indicate scalars (*e.g.*, $x$). Concatenation of $n$ scalars is denoted as $[x]_{i \in \mathcal{I}_n}^{\oplus}$.

**Few-shot multi-scenario multi-view learning.** Using a few-shot learning framework for anomaly detection is not an entirely new concept. One of the most widely recognized methods in this domain is the Few-shot Scene-adaptive Anomaly Detection (FSAD) model [46]. This model leverages the meta-learning framework [23] to train a model using video data collected from various camera views within the same scenario[6], such as a university street. The model is then fine-tuned on a different camera viewpoint within the university site. Although the trained model can be adapted to novel viewpoints, still its adaptability is confined to the university scenario. To overcome this limitation, we introduce the Scenario-Adaptive Anomaly Detection (SA$^2$D) method. Diverging from existing

---

[6]In this paper, the term 'scenario' is employed to signify different contexts such as retail, manufacturing, the education sector, smart cities, and others. In the context of different camera views, existing literature sometimes uses the term 'scene' to refer to the camera scene from a specific camera viewpoint.

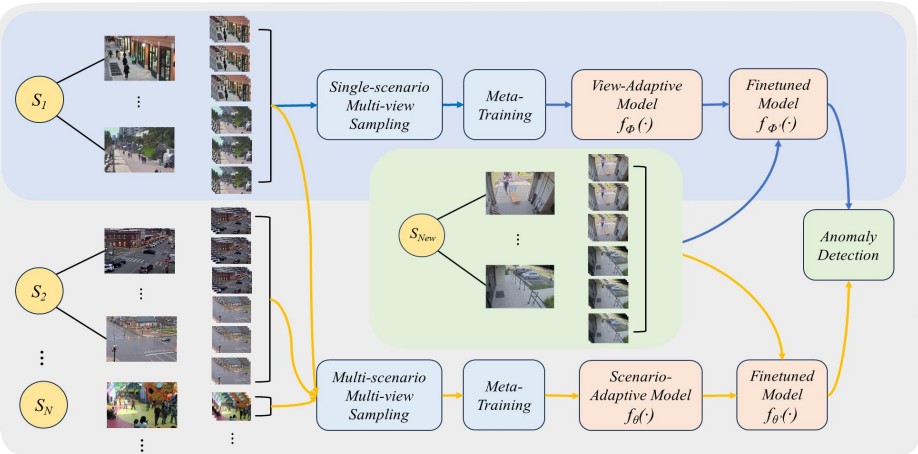

Figure 6: A comparison between the existing few-shot scene-adaptive (view-adaptive) anomaly detection model (depicted by the light blue block) and our proposed Scenario-Adaptive Anomaly Detection (SA$^2$D) model (illustrated by the light gray block). On the left-hand side of the figure, the first column, $\{S_1, S_2, \cdots, S_N\}$, represents different scenarios. The second column signifies various camera viewpoints, while the third column indicates the videos captured under each camera viewpoint. In contrast to the existing view-adaptive model, SA$^2$D extends its capabilities by incorporating a few-shot multi-scenario multi-view learning framework. The blue arrow illustrates the workflow of existing models, while the orange arrows show the workflow of our model.

few-shot anomaly detection models [46], our approach: (i) builds on and extends one-scenario multi-view to multi-scenario multi-view learning problem, and (ii) broadens the scope of test cases from multiple camera views to encompass novel scenarios, as well as multiple camera views per scenario. Our model stands out as a more advanced and versatile version in comparison to existing anomaly detection models. Fig. 6 shows a comparison between the existing few-shot anomaly detection model and our SA$^2$D model.

Our model operates at the frame level, recognizing the significance of early anomaly detection, where swift action is imperative to prevent or mitigate potential issues. We opt for a future frame prediction model, employing a $T$-frame video $\mathbf{V} = [\boldsymbol{F}_1, \boldsymbol{F}_2, \cdots, \boldsymbol{F}_T] \in \mathbb{R}^{H \times W \times T}$. For the sake of simplicity, we omit the three color channels. A temporal sliding window of size $T'$ with a step size of 1 is used to select video sub-sequences (termed video temporal blocks) $\mathbf{V}_i = [\boldsymbol{F}_i, \boldsymbol{F}_{i+1}, \cdots, \boldsymbol{F}_{i+T'-1}] \in \mathbb{R}^{H \times W \times T'}$, where $i$ represents the $i$th temporal block. For simplicity, we omit $i$ in the subsequent discussion, e.g., $\mathbf{V} = [\boldsymbol{F}_1, \boldsymbol{F}_2, \cdots, \boldsymbol{F}_{T'}] \in \mathbb{R}^{H \times W \times T'}$ denotes a temporal block. We consider the first $T'-1$ frames as the input $\mathbf{X} = [\boldsymbol{F}_1, \boldsymbol{F}_2, \cdots, \boldsymbol{F}_{T'-1}] \in \mathbb{R}^{H \times W \times (T'-1)}$ to future frame prediction model, and the last frame as the output $\boldsymbol{Y} = \boldsymbol{F}_{T'} \in \mathbb{R}^{H \times W}$, forming an input/output pair $(\mathbf{X}, \boldsymbol{Y})$. Our future frame prediction model is represented as $f_\theta : \mathbf{X} \to \boldsymbol{Y}$.

**Sampling strategy.** In the context of meta-learning, we begin by sampling a set of $N$ scenarios $\{S_1, S_2, \cdots, S_N\}$. Under each scenario, we randomly select one camera view from a set of $M$ camera viewpoints $\{V_1, V_2, \cdots, V_M\}$. Under the chosen camera view, we sample $K$ video temporal blocks to formulate an $N$-way $K$-shot learning problem. This sampling strategy enables the creation of a corresponding task $\mathcal{T}_{n,m} = \{\mathcal{D}_{n,m}^{\mathrm{tr}}, \mathcal{D}_{n,m}^{\mathrm{val}}\}$ per training episode, where $n \in \mathcal{I}_N$, $m \in \mathcal{I}_M$, and $\mathcal{D}_{n,m}^{\mathrm{tr}} = \{(\mathbf{X}_1, \boldsymbol{Y}_1), (\mathbf{X}_2, \boldsymbol{Y}_2), \cdots, (\mathbf{X}_K, \boldsymbol{Y}_K)\}$. These $K$ pairs of $\mathcal{D}_{n,m}^{\mathrm{tr}}$ are utilized to train the future frame prediction model $f_\theta$. Additionally, we sample a subset of input/output pairs to form the validation set $\mathcal{D}_{n,m}^{\mathrm{val}}$, excluding those pairs in $\mathcal{D}_{n,m}^{\mathrm{tr}}$. For the backbone selection, we adopt U-Net [76] for future frame prediction, incorporating a ConvLSTM block [83] for sequential modeling to retain long-term temporal information. Following [43] and [46, 113], we incorporate GAN [27] architecture for video frame reconstruction. It is important to note that the backbone architectures are not the primary focus of our work, and in theory, any anomaly detection network can serve as the backbone architecture. Given that each training episode randomly selects the camera viewpoint per scenario, and the scenarios are also randomly selected, our training scheme can be viewed as a few-shot multi-scenario multi-view learning.

**Training.** Given a pretrained anomaly detection model $f_\theta$, following [23, 46], we define a task $\mathcal{T}_{n,m}$ by establishing a loss function on the training set $\mathcal{D}_{n,m}^{\text{tr}}$ of this task:

$$\mathcal{L}_{\mathcal{T}_{n,m}}(f_\theta; \mathcal{D}_{n,m}^{\text{tr}}) = \sum_{(\mathbf{X}_{n,m}, \boldsymbol{Y}_{n,m}) \in \mathcal{D}_{n,m}^{\text{tr}}} L(f_\theta(\mathbf{X}_{n,m}), \boldsymbol{Y}_{n,m}), \tag{1}$$

Here, $L(f_\theta(\mathbf{X}_{n,m}), \boldsymbol{Y}_{n,m})$ computes the difference between the predicted frame $f_\theta(\mathbf{X}_{n,m})$ and the ground truth frame $\boldsymbol{Y}_{n,m}$. Following [46], we define $L(\cdot)$ as the summation of the least absolute deviation ($L_1$ loss) [62], multi-scale structural similarity measurement [112], and the gradient difference loss [52]. The updated model parameters $\theta'$ are adapted to the task $\mathcal{T}_{n,m}$. On the validation set $\mathcal{D}_{n,m}^{\text{val}}$, we measure the performance of $f_{\theta'}$ as:

$$\mathcal{L}_{\mathcal{T}_{n,m}}(f_{\theta'}; \mathcal{D}_{n,m}^{\text{val}}) = \sum_{(\mathbf{X}_{n,m}, \boldsymbol{Y}_{n,m}) \in \mathcal{D}_{n,m}^{\text{val}}} L(f_{\theta'}(\mathbf{X}_{n,m}), \boldsymbol{Y}_{n,m}), \tag{2}$$

We formally define the objective of meta-learning as:

$$\min_\theta \sum_{\substack{n \in \mathcal{I}_N, \\ m \in \mathcal{I}_M}} \mathcal{L}_{\mathcal{T}_{n,m}}(f_{\theta'}; \mathcal{D}_{n,m}^{\text{val}}). \tag{3}$$

where $N$ and $M$ denotes respectively the number of scenarios and camera views. Note that Eq. (3) sums over all tasks during meta-training. In practice, we sample a mini-batch of tasks per iteration.

**Inference.** During the meta-testing stage, when provided with a video from a specific camera view of a new scenario or a novel camera viewpoint of a known scenario $S_{\text{new}}$, we employ the first several frames of the video in $S_{\text{new}}$ for adaptation (using Eq. (1)) and then utilize the rest of the frames for testing.

**Scoring.** For anomaly detection, we calculate the disparity between the predicted frame $\hat{\boldsymbol{F}}_t$ and the ground truth frame $\boldsymbol{F}_t$, $t \in \mathcal{I}_T$. Following [43], we use the Peak Signal-to-Noise Ratio (PSNR) as a metric to evaluate the quality of the predicted frame:

$$\text{PSNR}(\boldsymbol{F}_t, \hat{\boldsymbol{F}}_t) = 10 \log_{10} \frac{(\max \boldsymbol{F}_t)^2}{\frac{1}{HW} \sum_{i=1}^H \sum_{j=1}^W (\boldsymbol{F}_t[i,j] - \hat{\boldsymbol{F}}_t[i,j])^2}, \tag{4}$$

here, $\max \boldsymbol{F}_t$ is the maximum possible pixel value of the ground truth frame, $[i,j]$ are the spatial location of the video frame, $H$ and $W$ are the height and width of the frame respectively. In general, higher PSNR values indicate better generated quality. The PSNR value of a video frame can be used to assess the consistency of the frames, with higher PSNR values indicating normal events as the frame is well predicted. Following [52], we normalize the PSNR scores of a $T$-frame video for anomaly scoring:

$$s_t = \frac{\text{PSNR}(\boldsymbol{F}_t, \hat{\boldsymbol{F}}_t) - \min[\text{PSNR}(\boldsymbol{F}_\tau, \hat{\boldsymbol{F}}_\tau)]_{\tau \in \mathcal{I}_T}^\oplus}{\max[\text{PSNR}(\boldsymbol{F}_\tau, \hat{\boldsymbol{F}}_\tau)]_{\tau \in \mathcal{I}_T}^\oplus - \min[\text{PSNR}(\boldsymbol{F}_\tau, \hat{\boldsymbol{F}}_\tau)]_{\tau \in \mathcal{I}_T}^\oplus}, \tag{5}$$

where $s_t$ varies within the range of 0 to 1. The normalized PSNR value can be used to assess the abnormality of a specific frame. A predefined threshold, *e.g.*, 0.8, can be set to determine whether a specific frame, such as $\boldsymbol{F}_t$, is considered an anomaly or not, by comparing it with $s_t$.

## E   Further discussions

**Advances in feature learning.** Multiple Instance Learning (MIL) framework is introduced in [85]. They treat normal and abnormal videos as bags, short clips of each video as instances, and use a ranking loss to distinguish between the highest-scoring abnormal and normal instances. Researchers [22] notice that directly using pre-trained models as feature extractors may not be suitable for surveillance as they are pre-trained on action recognition datasets. Moreover, the snippet with the highest anomaly score introduced in [85] may not come from an abnormal snippet within an abnormal video [90]. Their advanced work proposes a theoretical model to better differentiate between the top-$k$ snippet feature magnitudes of abnormal and normal snippets. However, relying solely on

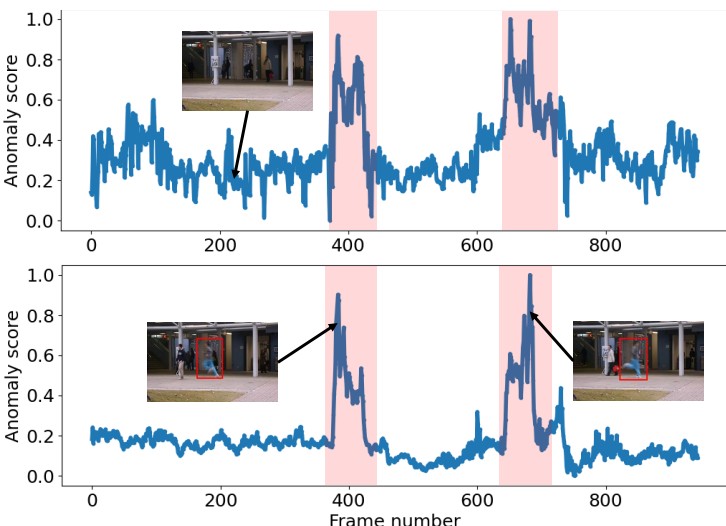

Figure 7: Frame-level anomaly scores (normalised between 0 and 1) are depicted for a test video from CUHK Avenue. The light red color block highlights the time period when the anomaly occurs. The top row illustrates the curve generated by the view-adaptive model, while the bottom row displays our scenario-adaptive model. Notably, our SA$^2$D exhibits considerably smoother curves compared to the few-shot scene-adaptive model. This observation suggests that our model, built on our MSAD, demonstrates superior anomaly detection performance.

Table 6: A comparison of recent state-of-the-art self-supervised and weakly-supervised methods on five benchmarks: UCSD Ped2 (Ped2), CUHK Avenue (CUHK), ShanghaiTech (ShT), UCF-Crime (UCF), and XD-Violence (XD). For Ped2, CUHK, ShT, and UCF, frame-level Micro-AUC (%) is used for evaluation, while for XD, average precision (%) is used.

| Paradigm | Year | Method | Description | Modality | Dataset | | | | |
|---|---|---|---|---|---|---|---|---|---|
| | | | | | Ped2 | CUHK | ShT | UCF | XD |
| **Self-sup.** | 2018 | Liu *et al.* [43] | Future frame prediction | RGB+opt. | 95.4 | 85.1 | 72.8 | - | - |
| | 2019 | Ionescu *et al.* [33] | Object-centric encoder | RGB | 97.8 | 90.4 | 84.9 | - | - |
| | 2019 | MemAE [26] | Memory module | RGB | 94.1 | 83.3 | 71.2 | - | - |
| | 2020 | FSAD [46] | Few-shot scene adaptive | RGB | 96.2 | 85.8 | 77.9 | - | - |
| | 2020 | MNAD [61] | Memory module | RGB | 97.0 | 88.5 | 70.5 | - | - |
| | 2021 | Georgescu *et al.* [25] | Background agnostic method | RGB | 98.7 | 92.3 | 82.7 | - | - |
| | 2021 | MPN [49] | Meta prototype unit | RGB | 96.9 | 89.5 | 73.8 | - | - |
| | 2021 | AEP [124] | Adversarial learning | RGB | 97.3 | 90.2 | - | - | - |
| | 2022 | Wang *et al.* [95] | Spatio-temporal jigsaw puzzles | RGB | 99.0 | 92.2 | 84.3 | - | - |
| | 2023 | STG-NF [31] | Normalizing flow | Skeleton | - | - | 85.9 | - | - |
| | 2023 | FPDM [118] | Diffusion model | RGB | - | 90.1 | 78.6 | 74.7 | - |
| | 2023 | Ristea *et al.* [72] | Lightweight masked encoder | RGB | 95.4 | 91.3 | 79.1 | - | - |
| | 2023 | HSC [87] | Semantic contrastive learning | RGB | 98.1 | 93.7 | 83.4 | - | - |
| | 2023 | USTN-DSC [120] | Video event restoration | RGB | 98.1 | 89.9 | 73.8 | - | - |
| **Weakly-sup.** | 2018 | Sultani *et al.* [85] | Multiple instance learning | RGB | - | - | - | 75.4 | - |
| | 2020 | Wu *et al.* [114] | Multimodal fusion | RGB+audio | - | - | - | - | 78.6 |
| | 2021 | MIST [22] | Task-specific encoder | RGB | - | - | 94.8 | 82.3 | - |
| | 2021 | RTFM [90] | Feature magnitude learning | RGB | 98.6 | - | 97.2 | 84.3 | 77.8 |
| | 2022 | MSL [40] | Multi-sequence learning | RGB | - | - | 97.3 | 85.6 | 78.6 |
| | 2022 | UBnormal [2] | Synthesis dataset | RGB | - | 93.0 | 83.7 | - | - |
| | 2023 | MGFN [14] | Glance-and-focus | RGB | - | - | - | 87.0 | 80.1 |
| | 2023 | UR-DMU [129] | Dual-memory units | RGB | - | - | - | 87.0 | 81.7 |
| | 2023 | VAD-CLIP [116] | Vision-language model | RGB+text | - | - | - | 88.0 | 84.5 |
| | 2023 | TEVAD [13] | Multimodal fusion | RGB+text | 98.7 | - | 98.1 | 84.9 | 79.8 |
| | 2023 | Pu *et al.* [64] | Prompt-enhanced learning | RGB+text | - | - | 98.1 | 86.8 | 85.6 |
| | 2024 | ITC [42] | Anomaly text completion | RGB+text | - | - | - | 89.0 | 85.5 |

feature magnitude to distinguish between normal and abnormal snippets can be ambiguous in certain situations [14]. To address this issue, a magnitude contrastive loss is introduced in [14] which enhances the similarity of feature magnitudes for videos within the same category and increases the separability between normal and abnormal videos. In comparison to earlier works [90, 14], Zhou *et al*. [129] emphasize the shortcomings of learning discriminative features for anomalous instances while neglecting the implication of normal data. Inspired by the methods with memory banks [4, 26, 33], they propose dual memory units for recording both normal and abnormal patterns.

**Enhancing anomaly understanding.** Most existing methods primarily focus on detection accuracy [60], while only a few works address interpretable anomaly detection [18, 70]. Since anomalous behaviors are scene-dependent [6], it is essential for detection methods to understand the underlying rules and constraints of the scene in order to comprehend the causes and consequences of anomalies. Although some existing works attempt to integrate visual and semantic information to understand anomalies [13, 64, 116, 126], the guidance provided by semantic information for anomaly detection is still insufficient. Therefore, continuing to explore new methods, *e.g*., Large Language Models (LLMs) [18, 50, 127], that enable models to better understand the semantic information of anomalies is crucial for advancing the field.

**Anomaly anticipation.** Most existing methods prioritize detecting anomalies rather than predicting them. Recently, Cao *et al*. [6] introduce the concept of video anomaly anticipation, which aims to predict anomalies in advance to enable early warnings [21]. This proactive approach leverages trends or clues in the events to foresee potential incidents, allowing timely intervention to prevent accidents. For example, early detection of smoke before a fire ignites, masked suspects wielding knives, or individuals about to fall can mitigate risks and prevent anomalies from happening. This forward-looking strategy not only improves anomaly detection systems but also stimulates research and in predictive analytics and real-time intervention methods.

**Lightweight models.** Despite the advent of powerful models in recent years, most existing methods consume substantial computational resources and runtime for extracting video features [59] or using an object detector [33, 24], creating a bottleneck for deployment in real-world scenarios [94, 59]. While a few studies have focused on model light-weighting [72], the trade-off between accuracy and model efficiency remains a significant challenge. To address these issues, several approaches can be considered. On one hand, data-efficiency methods such as data distillation can be employed to reduce redundant video data, thereby decreasing the computational cost [59]. On the other hand, improving the model architecture to learn more representative features or using knowledge distillation during deployment can enhance the efficiency. Furthermore, future research should consider incorporating the running time as an evaluation metric to better access the practical utility of these models in real-world scenarios.

**With vision-language models.** Exploring advanced deep learning and generative models to better capture the complexity of real-world scenarios is of vital importance. In particular, using pretrained large-scale vision-language models (VLMs) enhances the understanding of anomalies in videos and facilitates a more coherent reasoning process [119, 28, 50]. Notably, researchers have achieved state-of-the-art performance on multiple VAD tasks by using CLIP [66] to extract deep features, followed by a lightweight trainable network [116]. Modeling both visual and language information for multimodal fusion, contextual understanding, handling complex scenarios, and providing explanatory descriptions would be an interesting future direction.

**Quantitative evaluations.** Table 6 presents a comparison of recent self-supervised and weakly-supervised methods on five popular video anomaly detection datasets. We observe that UCSD Ped2 (Ped2), CUHK Avenue (CUHK), and ShanghaiTech (ShT) are widely used for self-supervised learning methods, whereas ShanghaiTech (ShT), UCF-Crime (UCF), and XD-Violence (XD) are commonly employed for weakly-supervised learning approaches. Additionally, we note that weakly-supervised methods generally outperform self-supervised methods, especially on more complex datasets such as ShanghaiTech (ShT). These findings suggest that incorporating additional weak supervision can significantly enhance the detection performance of VAD models, particularly in challenging environments.

Table 7: Performance evaluations by anomaly type (a total of 11 main anomaly types) on our MSAD test set are conducted. We use frame-level Micro AUC (%) and Average Precision (AP, in %) as evaluation metrics for models pretrained on either UCF-Crime or our MSAD. We use I3D as the backbone for all methods. The best training scheme for each method is highlighted in bold.

| Training set | Method | Assault | | Explosion | | Fighting | | Fire | |
|---|---|---|---|---|---|---|---|---|---|
| | | AUC | AP | AUC | AP | AUC | AP | AUC | AP |
| UCF-Crime | RTFM [90] | 60.6 | 62.2 | **69.3** | **79.0** | 68.5 | 80.7 | 36.0 | 64.5 |
| | MGFN [14] | **60.5** | **62.0** | **65.5** | **74.3** | 53.6 | 63.9 | 21.6 | 55.0 |
| | UR-DMU [129] | **59.4** | 60.5 | **69.3** | **82.0** | 71.2 | 85.2 | 36.2 | 66.5 |
| **MSAD** | RTFM [90] | **68.1** | **67.3** | 46.8 | 60.4 | **89.6** | **93.0** | **61.3** | **81.2** |
| | MGFN [14] | 59.7 | 59.0 | **64.5** | 71.9 | **89.4** | **93.5** | **86.0** | **93.0** |
| | UR-DMU [129] | 56.9 | **64.5** | 67.9 | 74.5 | **83.9** | **90.4** | **61.2** | **82.9** |

| Training set | Method | Object Falling | | People Falling | | Robbery | | Shooting | |
|---|---|---|---|---|---|---|---|---|---|
| | | AUC | AP | AUC | AP | AUC | AP | AUC | AP |
| UCF-Crime | RTFM [90] | 82.0 | 88.8 | **69.5** | **63.0** | **76.8** | **90.6** | 59.7 | 65.7 |
| | MGFN [14] | 65.5 | 73.1 | **57.2** | **59.5** | 72.0 | **89.1** | 42.1 | 57.6 |
| | UR-DMU [129] | 72.4 | 76.5 | **69.3** | **57.6** | **69.7** | **81.5** | 59.9 | 73.8 |
| **MSAD** | RTFM [90] | **94.7** | **96.7** | 56.5 | 50.4 | 65.7 | 81.2 | **78.2** | **84.7** |
| | MGFN [14] | **90.9** | **94.8** | 52.7 | 47.8 | **73.9** | 86.7 | **86.8** | **88.5** |
| | UR-DMU [129] | **92.1** | **95.8** | 42.5 | 43.7 | 63.5 | 79.3 | **81.4** | **87.8** |

| Training set | Method | Traffic Accident | | Vandalism | | Water Incident | | **Overall** | |
|---|---|---|---|---|---|---|---|---|---|
| | | AUC | AP | AUC | AP | AUC | AP | AUC | AP |
| UCF-Crime | RTFM [90] | 55.6 | 45.1 | **86.0** | **85.2** | 93.5 | 98.5 | 71.9 | 47.4 |
| | MGFN [14] | 52.6 | 45.3 | 80.7 | **81.4** | 41.0 | 81.7 | 61.8 | 31.2 |
| | UR-DMU [129] | 53.0 | 47.9 | **91.6** | **89.7** | 64.6 | 91.3 | 74.3 | 53.4 |
| **MSAD** | RTFM [90] | **62.2** | **51.8** | 85.2 | 76.1 | **96.3** | **99.1** | **86.7** | **66.3** |
| | MGFN [14] | **68.6** | **54.5** | **82.4** | 80.1 | **85.5** | **97.0** | **85.0** | **63.5** |
| | UR-DMU [129] | **62.0** | **55.6** | 84.7 | 77.0 | **98.5** | **99.5** | **85.0** | **68.3** |

# F   Further evaluations

**Scenario-adaptive outperforms view-adaptive.** We commence by comparing our SA$^2$D with the view-adaptive model (few-shot scene-adaptive [46]), utilizing test videos from CUHK Avenue for evaluation. In Fig. 7, we present the visualization of frame-level anomaly scores for both models. The illustration indicates that our SA$^2$D (Fig. 7 *bottom*) produces significantly smoother curves compared to the view-adaptive model (Fig. 7 *top*). This observation underscores our model's enhanced adaptability to new scenarios, affirming the superior performance of our scenario-adaptive approach over the view-adaptive model.

**Performance comparison by anomaly type.** Table 7 presents the frame-level Micro AUC and Average Precision (AP) performance, in percentage, for three weakly-supervised methods (RTFM [90], MGFN [14], and UR-DMU [129]) pretrained on either UCF-Crime or our MSAD training set. These methods are applied to our MSAD test set across various anomaly types (a total of 11 main anomaly types, as shown in Fig. 3a). We use I3D as the backbone for all methods in these experiments. We observe that methods pretrained on our MSAD dataset generally achieve better performance on anomalies such as fighting, fire, object falling, shooting, traffic accidents, and water incidents. Overall, using our dataset leads to better performance (see the overall performance column) compared to using UCF-Crime. This indicates that existing datasets do not focus sufficiently on non-human-related anomalies, and our dataset helps bridge this gap. However, for some anomaly types, such as explosions, people falling, and vandalism, using the UCF-Crime dataset achieves slightly better performance. The potential reasons for this are: (i) the I3D backbone encoder's difficulty in capturing sudden motions, as it is originally designed for action recognition, and (ii) limitations of our dataset, which suggests the need for further collection and augmentation of video samples to improve model performance. We also note that there is no single best performer for

Table 8: Performance evaluations by scenario (a total of 14 scenarios) on our MSAD test set are conducted. We use frame-level Micro AUC (%) and Average Precision (AP, in %) as evaluation metrics for models pretrained on either UCF-Crime or our MSAD. We use I3D as the backbone for all methods. The best training scheme for each method is highlighted in bold.

| Training set | Method | Frontdoor | | Highway | | Mall | | Office | |
|---|---|---|---|---|---|---|---|---|---|
| | | AUC | AP | AUC | AP | AUC | AP | AUC | AP |
| UCF-Crime | RTFM [90] | 80.8 | 80.1 | 37.1 | 1.4 | 86.0 | **87.1** | 68.5 | 63.2 |
| | MGFN [14] | 68.4 | 70.2 | 36.3 | 1.4 | **79.6** | **80.4** | 64.5 | 60.2 |
| | UR-DMU [129] | 84.7 | 82.6 | 18.9 | 1.1 | 83.1 | 80.6 | 66.6 | 57.6 |
| **MSAD** | RTFM [90] | **84.1** | **81.1** | **63.7** | **4.1** | **87.2** | 72.2 | **78.1** | **68.8** |
| | MGFN [14] | **86.4** | **85.1** | **79.7** | **4.1** | 65.3 | 56.6 | **75.1** | **62.4** |
| | UR-DMU [129] | **84.8** | **82.8** | **31.5** | **1.3** | **91.0** | **83.8** | **77.8** | **67.3** |

| Training set | Method | Park | | Parkinglot | | Pedestrian st. | | Restaurant | |
|---|---|---|---|---|---|---|---|---|---|
| | | AUC | AP | AUC | AP | AUC | AP | AUC | AP |
| UCF-Crime | RTFM [90] | **75.3** | 23.7 | 66.7 | 16.7 | 84.1 | **67.6** | 66.5 | 56.5 |
| | MGFN [14] | 55.3 | 7.9 | 59.5 | 12.3 | 74.4 | 11.2 | 47.3 | 32.4 |
| | UR-DMU [129] | **91.6** | 34.8 | 62.2 | 17.6 | 58.5 | 6.1 | 75.7 | 74.4 |
| **MSAD** | RTFM [90] | 69.0 | **25.6** | 74.4 | 35.9 | **97.4** | 50.6 | **96.1** | **91.9** |
| | MGFN [14] | **77.9** | **38.3** | 68.1 | 14.5 | **88.0** | **20.4** | **95.8** | **91.8** |
| | UR-DMU [129] | 87.8 | **36.2** | **91.4** | **53.9** | **81.9** | **11.5** | **93.1** | **87.4** |

| Training set | Method | Road | | Shop | | Sidewalk | | Street highview | |
|---|---|---|---|---|---|---|---|---|---|
| | | AUC | AP | AUC | AP | AUC | AP | AUC | AP |
| UCF-Crime | RTFM [90] | **82.9** | **47.1** | **85.1** | 68.5 | **89.1** | **66.1** | 82.6 | **35.9** |
| | MGFN [14] | 54.4 | 18.3 | 69.4 | 60.4 | 47.4 | 26.4 | 37.2 | 8.3 |
| | UR-DMU [129] | 49.5 | 26.6 | 78.8 | **66.5** | 68.0 | 55.9 | 62.0 | 23.0 |
| **MSAD** | RTFM [90] | 54.0 | 16.8 | 80.6 | **77.3** | 52.5 | 17.1 | 43.3 | 12.3 |
| | MGFN [14] | **77.9** | **49.7** | **84.9** | **77.2** | **85.5** | **62.3** | **87.6** | **40.7** |
| | UR-DMU [129] | **83.0** | **64.4** | **81.3** | 64.5 | **86.5** | **64.1** | **85.0** | **37.7** |

| Training set | Method | Train | | Warehouse | | **Overall** | |
|---|---|---|---|---|---|---|---|
| | | AUC | AP | AUC | AP | AUC | AP |
| UCF-Crime | RTFM [90] | 52.2 | **5.0** | 82.3 | **52.8** | 71.9 | 47.4 |
| | MGFN [14] | 39.8 | 2.1 | 55.4 | 18.3 | 61.8 | 31.2 |
| | UR-DMU [129] | 51.3 | 2.6 | **86.9** | 54.0 | 74.3 | 53.4 |
| **MSAD** | RTFM [90] | **66.9** | 3.9 | 69.5 | 37.4 | **86.7** | **66.3** |
| | MGFN [14] | **53.0** | **3.1** | **72.3** | **30.9** | **85.0** | **63.5** |
| | UR-DMU [129] | **59.0** | **3.1** | 81.2 | **59.1** | **85.0** | **68.3** |

all anomaly types, indicating that our dataset can support the VAD community in exploring and developing more robust anomaly detection algorithms.

**Performance comparison by scenario.** We now perform evaluations on scenario concepts, and the results are summarized in Table 8. As shown in the table, some scenarios, such as highways and trains, achieve quite low performance. The possible reasons for this are: (i) limited video data available for these scenarios, and (ii) these scenarios are challenging due to complex motions involving varying directions, speeds, different objects, human movements, *etc*. On average, the models pretrained on our dataset achieve better performance compared to those pretrained on UCF-Crime. This demonstrates that our dataset covers a diverse range of scenarios and could potentially facilitate multi-scenario anomaly detection, especially when the definition of anomalies is uncertain.

**Cross-dataset evaluation.** In real-world VAD systems, the model's ability to generalize to unseen scenarios is crucial. To evaluate this capability, we implement a zero-shot performance evaluation. Specifically, we select four state-of-the-art weakly supervised methods: RTFM [90], UR-DMU [129], MGFN [14], and TEVAD [13], and train them on multi-scenario datasets, either UCF-Crime or our MSAD, using Protocol ii. Subsequently, we evaluate their performance on single-scenario datasets,

Table 9: Comparison of cross-dataset results using four recent anomaly detection models with the I3D backbone. UCF, ShT, CUHK, and Ped2 denote UCF-Crime, ShanghaiTech, CUHK Avenue, and UCSD Ped2, respectively. Improvements from using models pre-trained on the MSAD dataset are highlighted in red, while performance drops are indicated in blue.

| Method | UCF→ShT | UCF→CUHK | UCF→Ped2 | MSAD→ShT | MSAD→CUHK | MSAD→Ped2 |
|---|---|---|---|---|---|---|
| RTFM [90] | 42.62 | 50.76 | 60.03 | 39.59 (↓3.03%) | 63.23 (↑12.47%) | 57.97 (↓2.06%) |
| UR-DMU [129] | 46.69 | 45.67 | 62.90 | 35.05 (↓11.64%) | 58.86 (↑13.19%) | 66.84 (↑3.94%) |
| MGFN [14] | 37.58 | 44.48 | 51.75 | 48.10 (↑10.52%) | 56.66 (↑12.18%) | 62.09 (↑10.34%) |
| TEVAD [13] | 59.34 | 43.39 | 36.96 | 45.27 (↓14.07%) | 64.82 (↑21.43%) | 62.56 (↑25.60%) |

such as ShanghaiTech, CUHK Avenue, and UCSD Ped2, without further training or fine-tuning. This approach allows us to test how well these pre-trained models adapt to new, previously unseen scenarios, which is essential for practical applications.

Following [90], we select the top three snippets with the largest feature magnitudes from both normal and abnormal videos to train a snippet classifier, using I3D as the backbone. The evaluation is carried out using the Micro AUC metric, with results recorded in Table 9.

As shown in the table, models pre-trained on our MSAD dataset generally have better zero-shot performance on the CUHK and UCSD Ped2 datasets. However, on ShanghaiTech, the performance is not as good. This issue may be due to the categorization of biking and driving as anomalies, which does not align with reality.

**Content differences between normal and abnormal videos.** In fact, most existing anomalous videos exhibit significant visual differences from normal videos in the test set, in some existing datasets such as ShanghaiTech. To mitigate this, when collecting our dataset, we split long videos from the same camera viewpoint into shorter clips to ensure that these videos are not influenced by visual differences between normal and abnormal scenes. This approach allows our dataset to be used for exploring real anomalies rather than global frame textures / contexts. As shown in Table 7 and 8, our dataset can evaluate model performance on different anomaly types independent of scenarios/contexts and on different scenarios independent of anomaly types. For example, the Object Falling anomalies happen under diverse scenes including the front door, office, road, *etc*. However, the model also gained good performance (above 90% AUC score), it can be concluded that the model focuses more on actions rather than spurious features.

Below we provide more insights on our new dataset. (i) Diverse video scenarios: Our training and testing videos are sourced from a wide range of scenarios, all featuring complex environments such as busy supermarkets/streets, empty parking lots/stores, and footage captured during different times of the day (morning/night). This diversity makes it challenging for the model to learn from trivial spurious features like global frame textures. (ii) Model performance in different scenarios: Tables 7 and 8 show that the current models struggle in certain scenarios (low average precision) and do not perform well across all types of anomalies. For instance, the detection of anomalies like object falling achieves good results (over 90% AUC) in diverse scenes, including the front door, office, and road. This indicates that the model relies more on motion rather than scene characteristics for judgment. However, in some scenarios, like a pedestrian street, while the mean AUC across three methods reaches 89.1%, the AP is only 27.5%. This highlights the significant challenge posed by the complex scenes (*e.g.*, crowded streets) in our dataset to current detection methods. (iii) Feature magnitude analysis: We analyze the feature magnitude extracted from our dataset using different backbones (*e.g.*, I3D, SwinTransformer) for each snippet (16 frames). In previous datasets (*e.g.*, UCSD Ped, CUHK Avenue, UCF-Crime), the snippet-level feature magnitude increases significantly during anomalies, while it remains low during normal situations due to simpler scenes. However, in our dataset, this assumption, using feature magnitude to differentiate normal from abnormal events, does not hold, as normal situations in complex environments (*e.g.*, busy streets) also exhibit significant and noisy changes in feature magnitude. We aim to use our MSAD dataset to advance anomaly detection in more challenging scenarios in future work.

**Concerns on spurious correlations in anomalous event detection.** We acknowledge the potential for a model to learn spurious correlations related to gender, race, or location when training on our dataset, especially in action classes like assault, fighting, and robbery, where faces and locations may

be visible. We recognize the risk that a black-box video anomaly detection model could inadvertently learn such correlations, leading to biased or unfair outcomes.

To address this, we propose the following recommendations and safeguards: (i) implementing additional anonymization techniques in the dataset, such as blurring or masking faces and identifiable features in the training data, with our blurred version of MSAD provided for researchers; (ii) using balanced sampling strategies and data augmentation techniques to ensure diverse representation across demographics and locations, minimizing the risk of unintended associations; (iii) incorporating post-training bias detection methods to identify learned spurious correlations, applying techniques such as fairness-aware learning [38] to mitigate these biases during training and evaluation; (iv) encouraging the use of explainable AI methods [17, 18] in anomaly detection to provide insights into the model's decision features, promoting transparency to identify and address spurious correlations early; and (v) proposing continuous monitoring and evaluation of the model on diverse, representative test sets to ensure it generalizes well and avoids biased behavior in different contexts.

In future, we plan to incorporate these recommendations more explicitly and explore further safeguards to ensure the ethical use of our dataset in training video anomaly detection models.

## G   A review of advances in datasets

**From single-view to multi-view.** Existing datasets fall into two main categories: single-view, such as UCSD Ped [93] and CUHK Avenue [45], and multi-view like ShanghaiTech [47] and UBnormal [2]. UCSD Ped [93] and CUHK Avenue [45] primarily originate from campus surveillance videos. ShanghaiTech [47], a pioneering multi-view dataset, has served as a benchmark for various methods. However, akin to other university datasets, it lacks object diversity and deviates from real-world scenarios regarding environment variations. UCF-Crime [85] is the first real-world dataset with multiple views. Despite encompassing diverse anomaly types, including abuse, explosions, stealing, and fighting, its quality is suboptimal due to monochrome video footage, low resolution, moving cameras, and redundancy. Still, most existing datasets focus on human-related anomalies.

**From single-scenario to multi-scenario.** Single-scenario datasets [6, 45, 93, 47, 75] are typically sourced from specific locations with one or multiple camera viewpoints, such as universities or urban streets, capturing routine activities and occasional anomalies. Early datasets, such as UCSD Ped [93] and CUHK Avenue [45], are collected from a single camera viewpoint at a university. Although single-scenario datasets are valuable, they lack diversity in scenarios, each with distinct backgrounds and anomalies. For instance, a parking lot typically has less movement compared to a crowded street, and a shopping mall is busier than a university avenue. Training a model exclusively on a single scenario limits its ability to generalize, as it only learns the specific features of that scenario and performs poorly when applied to new scenes [46]. To address the limited real-world diversity, multi-scenario datasets, such as UCF-Crime [85] and CUVA [18], have been collected from online video platforms (*e.g.*, YouTube), offering a wide range of scenarios. These variations in scenarios, with diverse weather and lighting conditions, greatly enhance the robustness and adaptability of anomaly detection models for real-world applications. UCF-Crime [85] has become the largest and most popular benchmark for weakly-supervised methods. The CUVA [18] dataset is another large-scale benchmark focused on the causation of video anomalies and is the first to include high-quality text descriptions of video content. This is designed to aid video large language models (VLMs) in understanding the cause and effect of anomalies. However, unlike UCF-Crime [85], which exclusively collects surveillance videos from websites, CUVA sources its videos from news outlets and handheld cameras. This results in significant viewpoint shifts that may affect performance in surveillance.

**The role of synthetic datasets.** As obtaining videos that contain anomalies is challenging in the real world, and while some datasets simulate anomalies, researchers have begun exploring the use of synthetic datasets to generate normal and abnormal videos. For example, UBnormal [2], generated using Cinema4D software, simulates diverse events with 268 training, 64 validation, and 211 testing samples. Importantly, UBnormal is often used to augment real-world datasets rather than for direct model training, thereby enriching the diversity of VAD tasks. While synthetic datasets can produce multiple scenarios with diverse motions, there are challenges: (i) simulating non-human-related anomalies is difficult, and (ii) the simulated motions may not accurately reflect real-world dynamics [51]. Hence, more real-world datasets are still required to train robust VAD models, even though synthetic datasets can alleviate the issue of insufficient training sources.

**Expanding anomaly types.** Most existing benchmark datasets focus on human-related anomaly detection. UCSD Ped, CUHK Avenue, and ShanghaiTech center on pedestrian activities, with anomaly types including biking, cart movement, loitering, running, throwing objects, chasing, brawling, sudden motions, *etc*. UCF-Crime and XD-Violence focus on crime and violence, with anomalies such as assault, burglary, robbery, fighting, and shooting. Recent datasets like CUVA and UBnormal introduce multiple scenarios and a broader range of anomaly types, extending beyond human behavior-related anomalies. These datasets include non-human-related anomalies such as smoking, fireworks, forest fires, *etc*. Most existing datasets overlook the detection of non-human-related anomalies, because humans are the main source of anomalies. Therefore, there is still a demand to develop comprehensive benchmarks and methods to advance VAD.

**Datasets are becoming closer to real-life scenarios.** Early datasets such as ShanghaiTech and UCSD Ped have some drawbacks, including low resolution, grayscale imagery, and questionable anomaly types. For instance, activities like cycling, running, or skating on the street may be considered normal in these datasets. While datasets like Street Scene [68] and IITB Corridor [75] have expanded diversity with more frames and anomaly events, they are still not representative enough for learning real-world normality. These datasets generally neglect scene-dependent anomalies (events that are normal in one scene but abnormal in another) [6]. To address these limitations, the new NWPU Campus dataset introduced in [6] contains 43 scenes (camera views). Additionally, the recent CUVA dataset introduced in [18] serves as a large-scale, comprehensive benchmark for understanding the causation of video anomalies. With high-quality annotations, this dataset advances prompt-based VAD approaches, bringing datasets closer to real-life scenarios. However, since most videos in this dataset come from news sources, they suffer from dynamic subtitles, frequent scene changes, *etc*. Fig. 1 shows a visual comparison of some VAD datasets. As shown in this comparison, the datasets are increasingly resembling real-life scenarios in terms of resolution, scenario complexity, anomaly types, and external variations.

# H   Limitation and future work

For evaluation criteria, we use frame-level AUC and average precision for temporal dimension, however, we also note that some benchmarks (like UCSD Ped, CUHK, *etc*.) have pixel-level annotations for spatial anomaly localizations. This kind of annotation is time-consuming and label intensive for large benchmarks. Looking ahead, we plan to enhance our dataset to support more comprehensive evaluations. This may include incorporating additional textual descriptions or adding bounding box annotations, which would enable our dataset to be used for spatial anomaly localization and causal reasoning. By extending our dataset in this manner, we aim to facilitate more advanced research in anomaly detection. Furthermore, we intend to integrate spatio-temporal localization-based metrics, such as region- and time-based detection scores, to further advance video anomaly detection by exploring fine-grained anomaly regions within videos. We plan to include this evaluation metric in a future version of our dataset.

We focus on weakly-supervised learning methods in this work. We will consider both parametric and non-parametric models in the future scope of our work. While recent methods have largely focused on parametric approaches, it is inspiring that earlier research works use non-parametric methods for anomaly detection. These non-parametric techniques often involve analyzing optical flow to detect motion changes [29, 54], offering an interpretable perspective on anomaly perception. In our future work, we plan to draw inspiration from these methods to evaluate the disorder in motion changes.

Although our dataset covers a wide range of scenarios and anomaly types, there are still more scenarios and anomaly types we need to consider for real-world applications. Some anomaly types, such as uncontrolled animals and sudden crowd gatherings, are still missing from our dataset. Our future work will focus on collecting more scenarios and anomaly types. Videos for industrial scenarios, like adherence to safety protocols such as wearing hard hats or operating equipment safely, are difficult to obtain due to privacy issues. We will explore alternative solutions for these anomalies. Moreover, we will continue refining our dataset for existing anomalies, such as no right turns at specific crossings, for traffic-related anomalies. Additionally, we will provide more modalities to further expand our dataset, such as audio, video captions, and human skeletons. Given recent advances, there is a greater focus on reusable large-scale pretrained models and multi-modal fusion for robust VAD.

# I Potential societal impact

Our MSAD dataset is designed to advance video anomaly detection and offers crucial support for enhancing community safety and security in surveillance applications.

To protect privacy, we have applied anonymization techniques (see Sec. A) to remove identifiable human information, such as faces, poses, and gaits. However, privacy remains a critical area for ongoing research, especially in sensitive environments like homes, hospitals, and care facilities, where it is essential to balance effective detection with robust privacy safeguards.

In the future, we aim to explore new data modalities, including audio and skeletal sequences that inherently lack identifiable information, to further improve video anomaly detection.

