# OpenReview forum: "Advancing Video Anomaly Detection: A Concise Review and a New Dataset"
_NeurIPS.cc/2024/Datasets_and_Benchmarks_Track — NeurIPS 2024 Track Datasets and Benchmarks Poster_

### Official Review · Reviewer_sasR · 2024-07-25
**Video Anomaly Detection Review and Dataset**

**Rating:** 9
**Confidence:** 3
**Correctness:** Yes
**Clarity:** Yes

**Review:**

The Review of the paper is as follows:

1. The authors have thoroughly addressed all aspects of this work, either within the paper or through a detailed description in the Appendix, which is commendable.
2. They have also outlined all potential future works related to this research.
3. Additionally, a detailed description of the SA2D model has been provided.
4. The authors have effectively highlighted the limitations of their work.
5. This dataset is novel in its approach, addressing various patterns and non-human anomalies. Additionally, the analysis is thorough, incorporating comparisons with other datasets and existing models.

**Strengths:**

1. I liked that the authors thoroughly reviewed prior works on VAD datasets and the available deep-learning models.
2. It is good that the authors have performed a thorough evaluation to explore the capabilities of their MSAD dataset and the model.
3. I also enjoyed the fact that they evaluated their model trained over their dataset (MSAD) in single as well as cross-scenario.
4. Their dataset covers all the real-world scenarios in terms of anomalies that are possible to occur.

**Additional Feedback:**

No

**Documentation:**

Yes

**Limitations:**

The authors have adequately addressed their work's limitations and highlighted the societal impact in terms of privacy, which they will consider in their future work.

**Opportunities For Improvement:**

Section 4.3: Why MSAD was sensitive to the choices of backbones? Can you highlight the reason for the same?

**Relation To Prior Work:**

It is very well discussed in the paper about how their work differs from previous contributions. The related work in Table 1 is highly appreciated.

**Summary And Contributions:**

The authors of this paper emphasize the need for a new Video Anomaly Detection (VAD) dataset by conducting a comprehensive survey of existing related works. They introduce the proposed dataset as Multi-Scenario Anomaly Detection (MSAD). This dataset includes previously uncovered patterns such as motion patterns and challenging variations like different lighting and weather conditions, which prior datasets lack. MSAD comprises 35 human-related anomalies and 20 non-human anomalies. The authors analyze MSAD using recent deep learning models for VAD across diverse and evolving surveillance scenarios, specifically evaluating it with the UCF-Crime and ShanghaiTech datasets using three popular backbones. They discovered that MSAD's performance is sensitive to the choice of backbones for certain methods.

---

> ### Author Rebuttal · Authors · 2024-08-17
>
> **To Reviewer sasR:**
>
> **Why is MSAD sensitive to the choice of backbones?**
>
> Existing backbones, such as C3D, I3D, and SwinTransformer, have demonstrated strong performance in video anomaly detection on current datasets (see Table 4 for results on UCF-Crime and ShanghaiTech). However, these datasets have significant limitations: (i) they cover a very limited range of scenarios, motion dynamics, and anomaly types, and (ii) most anomaly detection methods rely on action recognition pre-trained models as backbones for feature extraction, particularly for human-related anomalies.
>
> As a result, these backbones tend to perform well because the anomaly detection datasets predominantly feature human-related motions. In contrast, our dataset introduces more challenges in terms of scenarios, camera viewpoints, and motion dynamics, encompassing both human and non-human-related motions. Therefore, our dataset is more sensitive to the choice of backbones in methods such as UR-DMU and MGFN.
>
> Our dataset can be used for selecting appropriate model backbones or exploring more powerful backbone networks that do not overly depend on existing action recognition models. Additionally, our dataset advances video anomaly detection by considering a wider range of scenarios and a broader spectrum of anomaly types.

---

### Official Review · Reviewer_wmDs · 2024-07-25
**Major contribution: a new dataset MSAD.**

**Rating:** 5
**Confidence:** 5
**Correctness:** Yes.
**Clarity:** Yes.

**Review:**

pros:
1. The paper is well-written and easy to follow, with the motivation for the new dataset clearly presented.
2. The dataset includes more diverse and challenging scenes, even with various weather and illumination conditions. This provides a solid foundation for developing video anomaly detection models.

Pros:
1. The paper devotes a lot of space to justifying the need for a new dataset, which could be condensed. Consequently, limited space is left for dataset description and experimentation presentation and discussion.
2. Giving the new datasets and experiments, it is better to include insightful discussion on the limitations of current methods.
3. No discussion on the selection on the evaluated methods and their implementation details.

**Strengths:**

Video anomaly detection is a crucial task in computer vision with significant real-world applications. This paper introduces a new video anomaly detection dataset featuring more diverse and complex scenes, offering a robust foundation for training superior models.

**Additional Feedback:**

Post-rebuttal review: After considering the authors' responses and the feedback from other reviewers, I still have concerns about the substantial content differences between normal and abnormal videos. These differences may make it easier for the models to distinguish between anomalous and normal videos. As a result, I will maintain my original rating.

**Documentation:**

Sufficient documentation on the dataset.

**Ethics:**

No ethical conerns.

**Limitations:**

Yes.

**Opportunities For Improvement:**

The manuscirpt devotes a lot of space to justifying the need for a new dataset, which could be condensed. It would be better to move the dataset curation process and the introduction of the dataset from the appendix to the main paper. Additionally, there should be more discussion and explanation about the choice of evaluation methods and the limitations of these methods when applied to the new dataset.

**Relation To Prior Work:**

Yes. Comparison to existing video anomaly detection datasets is presented in the manuscript.

**Summary And Contributions:**

This paper presents a very brief review on video anomaly detection and a new dataset MSAD.  Particularly, the dataset includes 14 distinct scenarios from various camera views, featuring diverse motion patterns and challenging variations in lighting and weather. Experiments with exiting methods are conducted on the proposed datasets and shows that MSAD is more challenging than previous video anomaly detection benchmarks.

---

> ### Author Rebuttal · Authors · 2024-08-17
>
> **To reviewer wmDS:**
>
> **1.Condense the need for a new dataset.**
>
> Thank you for the constructive feedback. We have (i) relocated the dataset details, collection process, usage, and maintenance from the appendix to the main paper, and (ii) moved Section 2.2 from the main paper to the appendix.
>
> **2.Insightful discussion on the limitations of current methods on our new dataset.**
>
> Below are some key insights.
>
> **(i) I3D vs. SwinTransformer:** Using I3D as a backbone generally outperforms SwinTransformer. Although SwinTransformer excels in capturing local and global spatial features, its reliance on attention mechanisms for temporal modeling may not be as finely tuned to detect subtle anomalies. I3D, designed to maintain temporal consistency across frames, is better suited for detecting anomalies that are temporally localized, such as sudden object appearances or unexpected behaviors. SwinTransformer's attention-based approach may lack the temporal coherence needed for such tasks. Additionally, I3D’s focus on motion and spatiotemporal features aligns well with anomaly detection requirements, especially in motion-based anomalies and subtle temporal variations present in our dataset.
>
> **(ii) No single best performer on MSAD:** RTFM (ICCV 2021) and TEVAD (CVPRW 2023) are more robust to backbone changes than UR-DMU (AAAI 2023) and MGFN (AAAI 2023). This suggests that existing methods may lack comprehensive evaluations across diverse scenarios. Our dataset provides a solid benchmark for training, testing, and evaluating superior models.
>
> **(iii) Improved generalizability:** Models trained on our dataset exhibit better generalizability and adaptability, particularly under few-shot settings (Table 2 in the main paper), showing superior performance on unseen camera viewpoints. Additionally, our SA$^2$D model, trained on MSAD, significantly boosts performance.
>
> **(iv) Rethinking anomaly detection:** Our dataset encourages rethinking the correctness and trustworthiness of anomaly detection methods. Cross-dataset evaluation (Table 9 in the Appendix) reveals that existing datasets may suffer from unrealistic anomaly definitions. For instance, models pre-trained on MSAD perform poorly on ShanghaiTech but better on CUHK and UCSD Ped2, highlighting the misalignment of anomaly categories like biking and driving with reality.
>
> **(v) Boosting existing methods:** Our dataset enhances the performance of existing anomaly detection methods (Table 7 in the Appendix). Models trained on MSAD consistently achieve the best performance on anomalies such as fighting, fire, object falling, shooting, traffic accidents, and water incidents, addressing the challenge of lacking a comprehensive benchmark dataset for model training.
>
> **(vi) Supporting multi-scenario anomaly detection:** Our dataset benefits multi-scenario anomaly detection, enabling systematic evaluation across multiple scenarios (Table 8 in the Appendix). Recent methods trained on MSAD generally achieve better performance across all 14 scenarios, demonstrating that existing works have primarily focused on single-scenario detection. Our dataset offers a robust foundation for training superior models for multi-scenario anomaly evaluation.
>
> These discussions have been added to our paper.

---

> ### Author Rebuttal · Authors · 2024-08-17
>
> **To reviewer wmDS:**
>
> **3.Discussion on the selection of evaluation methods.**
>
> We focus on weakly-supervised learning methods from a practical perspective, as discussed in Section 2.1 (‘Why Self-Supervised and Weakly-Supervised’) and Section 4.2 (‘Practical Applicability and Effectiveness’). The six methods we selected provide open-source implementations, allowing us to explore the impact of different backbone choices.
>
> Each method has its unique focus. MIST (CVPR 2021) introduces a multiple instance self-training framework to refine task-specific discriminative representations using only video-level annotations. RTFM (ICCV 2021) trains a feature magnitude learning function to identify positive instances, enhancing the robustness of the multiple instance learning approach. MSL (AAAI 2022) uses multi-sequence learning to predict both video-level anomaly probabilities and snippet-level anomaly scores. UR-DMU (AAAI 2023) focuses on learning representations of normal data while extracting discriminative features from abnormal data for a better understanding of normal states. MGFN (AAAI 2023) proposes a glance-and-focus network to amplify the discriminative power of feature magnitudes across different scenes. TEVAD (CVPRW 2023) uses both visual and text features to complement spatio-temporal features with semantic meanings of abnormal events.
>
> These methods enable us to conduct a fair and comprehensive evaluation across various aspects.
>
> **4.Implementation details**
>
> In line with the standard practice in state-of-the-art methods [1, 2, 3], we use C3D [4], I3D [5], and Video Swin Transformer [6], pretrained on Kinetics-400, as backbone networks for feature extraction. Although the same backbone is used, different methods may extract features of varying dimensions, which is not an apple-to-apple comparison. For example, UR-DMU [3] and MSL [2] extracts 1024-dimensional features, whereas other methods extract 2048-dimensional features when using the I3D model as the backbone.
>
> To ensure a fair comparison, following RTFM [1], we extract 4096D features from the ’fc6’ layer of the C3D network, 2048D, 10-crop features from the ’mix 5c’ layer of the I3D network, respectively. Additionally, following MSL, we extract 1024D features from the Video Swin Transformer from the stage 4 layer of the Video Swin Transformer network.We apply these feature extraction processes on three multi-scene datasets, including ShanghaiTech, UCF-Crime and our MSAD. Then we use these extracted features as inputs of different methods.
>
> **References:**
>
> [1] Tian, Yu, et al. "Weakly-supervised video anomaly detection with robust temporal feature magnitude learning." Proceedings of the IEEE/CVF international conference on computer vision. 2021.
>
> [2] Li, Shuo, Fang Liu, and Licheng Jiao. "Self-training multi-sequence learning with transformer for weakly supervised video anomaly detection." Proceedings of the AAAI Conference on Artificial Intelligence. Vol. 36. No. 2. 2022.
>
> [3] Zhou, Hang, Junqing Yu, and Wei Yang. "Dual memory units with uncertainty regulation for weakly supervised video anomaly detection." Proceedings of the AAAI Conference on Artificial Intelligence. Vol. 37. No. 3. 2023.
>
> [4] Github repository Pytorch_C3D_Feature_Extractor: https://github.com/yyuanad/Pytorch_C3D_Feature_Extractor
>
> [5] Github repository I3D_Feature_Extraction_resnet: https://github.com/GowthamGottimukkala/I3D_Feature_Extraction_resnet
>
> [6] Github repository Video-Swin-Transformer: https://github.com/SwinTransformer/Video-Swin-Transformer

---

> ### Comment · Reviewer_wmDs · 2024-08-27
> **Thanks for the rebuttal.**
>
> Thank you to the authors for addressing my previous concerns. After considering the authors' responses and the feedback from other reviewers, I still have concerns about the substantial content differences between normal and abnormal videos. These differences may make it easier for the models to distinguish between anomalous and normal videos. As a result, I will maintain my original rating.

---

> > ### Author Rebuttal · Authors · 2024-08-28
> >
> > **To reviewer wmDs:**
> >
> > Thank you for the prompt response and constructive feedback. Regarding your concerns about content differences between normal and abnormal videos, we would like to clarify the following:
> >
> > **1. Diverse video scenarios:** Our training and testing videos are sourced from a wide range of scenarios, all featuring complex environments such as busy supermarkets/streets, empty parking lots/stores, and footage captured during different times of the day (morning/night). This diversity makes it challenging for the model to learn from trivial spurious features like global frame textures.
> >
> > **2. Model performance in different scenarios:** Kindly refer to Tables 7 and 8 in our appendix, which show that the current model struggles in certain scenarios (low average precision) and does not perform well across all types of anomalies. For instance, the detection of anomalies like object falling achieves good results (over 90% AUC) in diverse scenes, including the front door, office, and road. This indicates that the model relies more on motion rather than scene characteristics for judgment. However, in some scenarios, like a pedestrian street, while the mean AUC across three methods reaches 89.1%, the AP is only 27.5%. This highlights the significant challenge posed by the complex scenes (e.g., crowded streets) in our dataset to current detection methods.
> >
> > **3. Feature magnitude analysis:** We analyzed the feature magnitude extracted from our dataset using different backbones (e.g., I3D, Video Swin Transformer) for each snippet (16 frames). In previous datasets (e.g., UCSD Ped, CUHK Avenue, UCF-Crime), the snippet-level feature magnitude increases significantly during anomalies, while it remains low during normal situations due to simpler scenes. However, in our dataset, this assumption—using feature magnitude to differentiate normal from abnormal events—does not hold, as normal situations in complex environments (e.g., busy streets) also exhibit significant and noisy changes in feature magnitude. We aim to use our MSAD dataset to advance anomaly detection in more challenging scenarios in future work.

---

### Official Review · Reviewer_veG2 · 2024-08-01
**A strong effort in improving VAD benchmarking, but needs significant improvement in review section and benchmark quality assessment**

**Rating:** 5
**Confidence:** 5
**Correctness:** 1) I mentioned my concern regarding b…

**Review:**

1). The paper is for most parts well-written and explained.

2). From the point of view of providing a comprehensive review of the field, I think the paper does not do a good job. The discussions provided in the paper regarding previous benchmarks and models mostly read like an extensive related work section. The review mostly focuses on the negative aspects of previous benchmarks and models to mainly highlight the advantages of their proposed dataset and model, which I think is not the correct way to present a review. Specifically, from the perspective of benchmarks, the paper fails to accurately define 'anomaly' in different settings (novel objects or actions, similar objects/actions in different scene locations/scene-context) and the utility of single scene benchmarks for evaluation of VAD methods. From the perspective of VAD methods, the paper mainly focuses on the representation learning component but does not provide information concerning model-building (For example, parametric vs non-parametric approaches, probabilistic methods like Bayesian Networks, etc). Even for representation learning, the paper fails to highlight the utility of learned interpretable features like object classes, and motion attributes and differentiate between spatial/temporal/spatio-temporal and semantic representations.  While I understand that an extensive review is not possible, I would still expect a review of the field to include the above information presented in an objective manner, rather than an overly-critical subjective presentation.

3). The proposed dataset provides a more challenging benchmark for evaluating video anomaly detection methods. Given the performance of most standard  VAD benchmarks (UCSD Ped2, CUHK, ShanghaiTech) is close to saturation, having a new benchmark with a larger train/test set, high-resolution videos, and videos with varied scene-context and low-level visual attributes is good for benchmarking VAD methods.

4). The proposed experimental setup is mostly correct.

5). The proposed method (scenario-adaptive anomaly detection) is intuitive and makes sense from the perspective of the overall motivation of the paper.

**Strengths:**

1) As mentioned in the Review section, the proposed benchmark is a strong step towards providing a more challenging benchmark for anomaly detection in videos based on actions/spatio-temporal events.

2) The proposed dataset is shown to be a good dataset for pre-training models, which can then be used on smaller benchmarks.

3) The proposed model incorporating multi-scenario data is shown to generate strong results.

**Additional Feedback:**

N/A

**Clarity:**

The paper is well written for the most parts. My only complaint would be that a lot of details regarding previous benchmark analysis including the taxonomy of evaluation criteria should be present in the main paper. Furthermore, the review sections could be presented more objectively.

**Documentation:**

The paper provides the link to the dataset and general details regarding dataset construction. However, no details are provided regarding licensing or future maintenance plans. The authors did not provide an explicit datasheet [3] for a better assessment of the provided benchmark and baseline models. No details/codebase were provided for the reproducibility of benchmark results and the proposed model.


[3] Gebru, Timnit, et al. "Datasheets for datasets." Communications of the ACM 64.12 (2021): 86-92.

**Ethics:**

While the authors mention that they obtained permission from public video hosting websites (YouTube, Itemfix, and Bilibili) for dataset collection, no details were provided about the permission. Furthermore, while the authors do mention in the supplementary material under the potential societal impact that the dataset contains human identity information, I believe this is a big red flag. Specifically, the dataset considers action classes like 'assault', 'fighting', and 'robbery' as anomalous events. In almost all the videos in the above categories, the faces of humans are perfectly visible, and in some videos, the location is easily discernable. My concern is that if a black-box method is trained for video anomaly detection using the provided dataset, it can learn spurious correlations with respect to the gender/race/location of human subjects in the videos. The authors fail to provide any recommendations/safeguards against the above issues.

**Limitations:**

The authors tried to adequately address limitations and potential negative societal impact. However, I believe there are some major concerns that need to be addressed. I have pointed out most concerns in the 'Opportunities For Improvement' section and the 'Ethics' section. Regarding visible human subjects and location information, I believe it's important to hide these by blurring the required areas in the videos.

**Opportunities For Improvement:**

1) As mentioned in the 'Review' section above, the paper does not do a good job of providing a comprehensive review of the field objectively. A lot of details are either absent or not presented in a structured manner. A lot of important details are provided in the supplementary material which should have been discussed in the main paper.

2) One big concern I have regarding the benchmark is that most of the anomalous videos have significant visual differences from the normal videos considered in the test set. Most normal videos look similar to training videos with respect to global video properties. I am concerned that it might be easier for models to differentiate anomalous videos from normal videos in a test set by learning trivial spurious features like global frame textures etc and not by learning about actions.
This brings me to my next concern regarding evaluation metrics. For the proposed benchmark, the key criteria selected for evaluation is Frame-level AUC. In the case of single-scene benchmarks like CUHK Avenue or single-scenario benchmarks like ShanghaiTech Frame-level AUC still correlates with anomaly localization performance (although not accurately and can be misleading ). However here in the case of a Multi-scenario benchmark, because of the strong correlation of activities with scene-context, the model can learn the spurious correlation of scene information (and not the actual anomalous actions) and use that for prediction and still get high Frame-level AUC score. Thus it is important to also utilize Sptio-temporal localization-based metrics like Region/Temporal-based detection scores.

3) Regarding the proposed model, it is not clear if the scene label information is used as a weak supervision implicitly in the sampling phase. If that is the case, then it is not fair to compare Shanghaitech pretraining with MSAD pretraining as when using MSAD we are providing more information during training, and thus it's not an apples-to-apples comparison.

4) The authors did not provide any other recent baseline models that perform unsupervised video anomaly detection. The baseline comparison with the FSAD model does not qualify as an unsupervised model as it has access to camera viewpoint information during model training.

5). The benchmark consists of human identities where faces are not blurred. I am concerned that this might violate the ethics criteria.

**Relation To Prior Work:**

The paper sufficiently discusses prior work.

**Summary And Contributions:**

The paper presents a short review and a new benchmark dataset for the task of Video Anomaly Detection. The authors first summarize the research conducted in the area of video anomaly detection from the perspective of dataset evolution and methodologies for anomaly detection focusing on representation learning and model learning settings (few-shot and weakly supervised learning). Specifically, the paper discusses the shortcomings of existing datasets with respect to the diversity of anomalous actions and scene context and similarly highlights the issues of context-sensitivity, generalizability, and adaptability with respect to existing models for anomaly detections.
The paper then proposes a new large-scale dataset comprising videos of multiple common-place scenarios under different settings like illumination, motion patterns, etc. The authors re-purpose an existing anomaly detection method [1] for multi-scenario settings and demonstrate strong performance when the model is trained on the new proposed dataset.

[1] Y. Lu, F. Yu, M. K. K. Reddy, and Y. Wang. Few-shot scene-adaptive anomaly detection. In Computer Vision–ECCV 2020: 16th European Conference, Glasgow, UK, August 23–28, 2020, Proceedings, Part V 16, pages 125–141. Springer, 2020.

---

> ### Author Rebuttal · Authors · 2024-08-17
>
> **To reviewer veG2:**
>
>
> **1.Important details should move from Appendix to the main paper.**
>
> Thank you for the constructive feedback.
>
> **Our review is objective.** Section 2.1 of the main paper discusses the evolution of the dataset, along with the methods developed in response to each introduced benchmark. We observe that each dataset and its corresponding state-of-the-art methods focus on addressing specific anomalies or scenarios.
>
> In Appendix G, we provided an additional review of advances in datasets. This section details the progression of video anomaly detection datasets, from single-view to multi-view, from single-scenario to multi-scenario, the role of synthetic datasets, and the efforts to expand anomaly types. These developments indicate that video anomaly detection datasets are becoming increasingly aligned with real-life scenarios to address various challenges.
>
> Our review identifies key practical issues in video anomaly detection, particularly the lack of comprehensive datasets featuring diverse scenarios.
>
> In response to this, we introduce the MSAD and evaluate six representative weakly-supervised learning methods to thoroughly assess their generalizability, adaptability, practical applicability, and effectiveness under both single-scenario and cross-scenario evaluation settings.
>
> We have now moved the dataset review from the appendix to the main paper. Additionally, we have: (i) Relocated the dataset details, collection process, usage, and maintenance from the appendix to the main paper, and (ii) Moved Section 2.2 from the main paper to the appendix.
>
> **Information regarding parametric vs. non-parametric.** Thank you for the suggestions. Our work focuses on weakly supervised learning methods in this work. We will consider both parametric and non-parametric models in the future scope of our work. While recent methods have largely focused on parametric approaches, it's inspiring that earlier research employed non-parametric methods for anomaly detection. These non-parametric techniques often involved analyzing optical flow to detect motion changes [6, 7], offering an interpretable perspective on anomaly perception. In our future work, we plan to draw inspiration from these methods to evaluate the disorder in motion changes.
>
> **Define anomaly.** Although there is no unified and clear definition of specific anomalies. Generally, we improved the definition of “anomaly” in the context of video anomaly detection, an "anomaly" refers to any event, behavior, or object in a video sequence that deviates from the normal or expected pattern of events. Given that anomalies are highly scene-dependent, single-scene datasets have a distinct advantage in testing a model’s performance under consistent, controlled conditions. This allows for a more focused evaluation of the model’s ability to detect deviations within a specific context.
>
> In Table 6 of the supplementary material, we present some **recent self-supervised methods**. Since most self-supervised experiments are conducted on single-scenario datasets, it is challenging for us to perform experiments on single-scenario settings using the multi-scenario SA$^2$D method, and there is a lack of corresponding baselines for multi-scenario methods.
>
> Scene-adaptive in existing literature ideally adapts a pretrained model to different scenes; however, the lack of a comprehensive scenario-based dataset causes practical issue: existing methods mostly address single scenario problem, e.g., viewpoint-adaptive under a specific scenario (ShanghaiTech focuses on a university scenario). Our scenario-adaptive, on the other hand, considers multiple scenarios.
>
> **2. Significant visual differences from the normal videos and concerns regarding evaluation metric.**
>
> In fact, most existing anomalous videos exhibit significant visual differences from normal videos in the test set, in some existing datasets such as ShanghaiTech.
>
> To mitigate this, when collecting our dataset, we split long videos from the same camera viewpoint into shorter clips to ensure that these videos are not influenced by visual differences between normal and abnormal scenes. This approach allows our dataset to be used for exploring real anomalies rather than global frame textures / contexts. As shown in Tables 7 and 8, our dataset can evaluate model performance on different anomaly types independent of scenarios/contexts and on different scenarios independent of anomaly types.
>
> For example, the Object Falling anomalies happen under diverse scenes including the front door, office, road, etc. However, the model also gained good performance (above 90% AUC score), it can be concluded that the model focuses more on actions rather than spurious features.
>
> For evaluation criteria, we use frame-level AUC and average precision for temporal dimension, however, we also noted that some benchmarks (like UCSD Ped, CUHK, etc.) have pixel-level annotations for spatial anomaly localizations. This kind of annotation is time-consuming and label intensive for large benchmarks. Looking ahead, we plan to enhance our dataset to support more comprehensive evaluations. This may include incorporating additional textual descriptions or adding bounding box annotations, which would enable our dataset to be used for spatial anomaly localization and causal reasoning. By extending our dataset in this way, we aim to facilitate more advanced research in anomaly detection.
>
> We appreciate the reviewer’s suggestion regarding the use of spatio-temporal localization-based metrics, such as region/temporal-based detection scores.
>
> We agree that incorporating this evaluation metric would further advance video anomaly detection by exploring fine-grained anomaly regions within videos. We plan to add this evaluation metric in a future version of our dataset.

---

> ### Author Rebuttal · Authors · 2024-08-17
>
> **To reviewer veG2:**
>
> **3. Scene information is implicitly used as weak supervision during the sampling phase**
>
> The scene information provided is used in the few-shot sampling strategy, where each scenario is treated as a group for sampling purposes. This allows the model to learn information from multiple scenarios in a balanced manner (see Figure 6 in appendix).
>
> In our proposed model, SA$^2$D, scene information is used during the sampling process to form N-way K-shot learning.
>
> Table 2's last four columns compare (i) the performance without scene information (FSAD [1]) and (ii) the performance with scene information (our proposed model SA$^2$D) on our MSAD dataset (Note that both models were trained using MSAD). The primary difference between FSAD and SA$^2$D is that SA$^2$D leverages scene information during sampling to enhance the N-way K-shot learning framework. Therefore, our comparison is fair.
>
> As shown in the table, our sampling strategy (Appendix D) significantly boosts SA$^2$D's performance across all five test splits on different camera viewpoints in ShanghaiTech.
>
> **4.Provide recent unsupervised video anomaly detection models**
>
> In this work, we focus on weakly-supervised learning methods from a practical perspective, as discussed in Section 2.1 (‘Why Self-Supervised and Weakly-Supervised’) and Section 4.2 (‘Practical Applicability and Effectiveness’).
>
> However, we appreciate the reviewer's feedback. Due to limited time, we are unable to conduct large-scale evaluations on unsupervised learning methods, but we plan to include these in an enhanced version of our work. We have identified the following unsupervised video anomaly detection methods with open-source implementations, which we intend to explore further: [2, 3, 4, 9]
>
> The baseline comparison provided in the paper includes the following: generalizability and adaptability evaluation through few-shot learning setups in both single-scenario and cross-scenario experiments (Tables 2 and 3 in the main paper), practical applicability and effectiveness (Table 4 in the main paper), performance evaluation by anomaly type (Table 7 in the appendix), performance evaluation by scenario (Table 8 in the appendix), and cross-dataset evaluation (Table 9 in the appendix).
>
> **5.Human faces in the dataset need to be blurred**
>
> Since in surveillance video footage, the facial regions of individuals are often small and blurry, or the camera angles typically capture side or rear views, it is challenging to extract identifiable information about individuals.
>
> To eliminate any identifiable information that might be visible, we followed the reviewers' suggestions and applied an automatic blurring script [5] to anonymize all the faces and car licenses in the videos of the MSAD dataset, then we check the whole dataset manually to ensure that all the identity information is almost eliminated. The blurred videos have limited impact to the detection performance while reducing the invasion of privacy.
>
> We have provided the blurred version of our MSAD dataset in our dataset submission at the Google drive link [8].
>
> **6.Details regarding licensing and future maintenance plans and reproducibility of benchmark**
>
> We have provided details on "Dataset Usage and Maintenance" in Appendix C.
>
> To address concerns about deprecated videos, we will regularly monitor the sources of our dataset to ensure that the videos or data links remain valid. This process may involve automated scripts that periodically verify online links and alert us if any sources become inaccessible. If the original video source becomes unavailable, we will seek alternative sources or mirrors that host the same content. Maintaining a list of alternative sources will help preserve the dataset's integrity. However, if no alternative sources are found, we may consider removing the unavailable content from our dataset to ensure that it remains up-to-date and reliable.
>
> Additionally, we will update the experimental results on our dataset website if any videos become unavailable. We will also document the missing videos for transparency and fairness in future research comparisons.
>
> We will provide a detailed datasheet to facilitate a more comprehensive assessment of the benchmark and baselines. In the submission link [8], we have provided our script for downloading the videos, as well as extracted features from backbone networks such as I3D and SwinTransformer. We will also release our full evaluation framework to allow interested researchers to further explore our multi-scenario datasets in the project website.
> We believe that our efforts provide a solid foundation for future research on real-world anomaly detection, ultimately benefiting our community.

---

> ### Author Rebuttal · Authors · 2024-08-17
>
> **To reviewer veG2:**
>
> **7.Concerns on spurious correlations in anomalous event detection.**
>
> We thank the reviewer for raising the concern regarding the potential for a model to learn spurious correlations related to gender, race, or location when training on our dataset, especially in action classes like 'assault,' 'fighting,' and 'robbery,' where faces and locations may be visible.
>
> We acknowledge the risk that a black-box video anomaly detection model could inadvertently learn such correlations, leading to biased or unfair outcomes. To address this, we propose the following recommendations and safeguards:
>
> (i) We recommend implementing additional anonymization techniques in the dataset, such as blurring or masking faces and identifiable features in the training data. We have provided our blurred version of MSAD for researchers.
>
> (ii) We suggest using balanced sampling strategies and data augmentation techniques to ensure diverse representation across different demographics and locations. This approach can help minimize the risk of the model forming unintended associations.
>
> (iii) Incorporating post-training bias detection methods can help identify any learned spurious correlations. Techniques such as fairness-aware learning [12] can be applied to mitigate these biases during model training and evaluation.
>
> (iv) Encouraging the use of explainable AI methods [10, 11] in anomaly detection can provide insights into what features the model is relying on for its decisions. This transparency can help identify and address any spurious correlations early in the process.
>
> (v) We propose continuous monitoring and evaluation of the model on diverse and representative test sets to ensure that it generalizes well and does not exhibit biased behavior in different contexts.
>
> In future versions of our work, we plan to incorporate these recommendations more explicitly and will explore further safeguards to ensure the ethical use of our dataset in training video anomaly detection models.
>
> **Reference:**
>
> [1] Lu, Yiwei, et al. "Few-shot scene-adaptive anomaly detection." Computer Vision–ECCV 2020: 16th European Conference, Glasgow, UK, August 23–28, 2020, Proceedings, Part V 16. Springer International Publishing, 2020.
>
> [2] Ristea, Nicolae-C., et al. "Self-distilled masked auto-encoders are efficient video anomaly detectors." Proceedings of the IEEE/CVF Conference on Computer Vision and Pattern Recognition. 2024.
>
> [3] Aich, Abhishek, Kuan-Chuan Peng, and Amit K. Roy-Chowdhury. "Cross-domain video anomaly detection without target domain adaptation." Proceedings of the IEEE/CVF Winter Conference on Applications of Computer Vision. 2023.
>
> [4] Yan, Cheng, et al. "Feature prediction diffusion model for video anomaly detection." Proceedings of the IEEE/CVF International Conference on Computer Vision. 2023.
>
> [5] The GitHub repository deface: https://github.com/ORB-HD/deface
>
> [6] Sharif, Md Haidar, and Chabane Djeraba. "An entropy approach for abnormal activities detection in video streams." Pattern recognition 45.7 (2012): 2543-2561.
>
> [7] Mehran, Ramin, Alexis Oyama, and Mubarak Shah. "Abnormal crowd behavior detection using social force model." 2009 IEEE conference on computer vision and pattern recognition. IEEE, 2009.
>
> [8] MSAD original dataset submission link for review (Google Drive):
> https://drive.google.com/drive/folders/1mxFGCcAuEecN0c7MHD12wD4pO2Ej5kLe
>
> [9] Hirschorn, Or, and Shai Avidan. "Normalizing flows for human pose anomaly detection." Proceedings of the IEEE/CVF International Conference on Computer Vision. 2023.
>
> [10] Doshi, Keval, and Yasin Yilmaz. "Towards interpretable video anomaly detection." Proceedings of the IEEE/CVF Winter Conference on Applications of Computer Vision. 2023.
>
> [11] Du, Hang, et al. "Uncovering What Why and How: A Comprehensive Benchmark for Causation Understanding of Video Anomaly." Proceedings of the IEEE/CVF Conference on Computer Vision and Pattern Recognition. 2024.
>
> [12] Le Quy, Tai, et al. "A survey on datasets for fairness‐aware machine learning." Wiley Interdisciplinary Reviews: Data Mining and Knowledge Discovery 12.3 (2022): e1452.

---

### Author Rebuttal · Authors · 2024-08-17

**To all reviewers:**

We would like to thank the reviewer for the constructive comments regarding the ethical considerations of our work. We have addressed the ethical concerns raised by the reviewer as outlined below. Please let us know if further clarification is needed.

While we acknowledge that surveillance videos can raise privacy and violence concerns, the potential risks are magnified when anomalies occur. A real-world anomaly detection dataset serves as a necessary complement, despite the inherent drawbacks of surveillance. However, it is crucial to implement appropriate regulations to minimize the potential negative impact of datasets.

Our MSAD dataset is intended exclusively for academic research. Researchers who wish to access the dataset must complete the online form [1] and agree to the terms and conditions [2]. We agree that ethical considerations are crucial when dealing with such data. The main ethical concerns raised by the reviewers are responded below:

**1. Identifiable information**

(i)   **Face Blurring:** Since in surveillance video footage, the facial regions of individuals are often small and blurry, or the camera angles typically capture side or rear views, it is challenging to extract identifiable information about individuals. To eliminate any identifiable information that might be visible, we followed the reviewers' suggestions and applied an automatic blurring script [3] to anonymize all the faces and car licenses in the videos of the MSAD dataset, then we check the whole dataset manually to ensure that all the identity information is almost eliminated. The blurred videos have limited impact to the detection performance while reducing the invasion of privacy. We have provided the blurred version of our MSAD dataset. Please check the original dataset link [12].

(ii)   **Extracted Features:** We have also released the extracted video features with different backbones (e.g., I3D, Video SwinTransformer) to the research community to promote privacy preservation. For extracted features, please check the original dataset link [12].  We have noticed that some weakly-supervised methods in recent years only use extracted video features as inputs [4, 5, 6, 7]. Training and evaluation with extracted video features is lightweight while protecting privacy.

(iii)  **Deepfake Technique:** We also explored using deepfake techniques to remove identifiable features from the original videos, aiming to preserve characteristics such as age and gender while minimizing information loss compared to face blurring. However, we encountered challenges due to the diverse resolutions and camera angles, which made deepfake difficult. We plan to further investigate this approach in future work.

**2. Ethical approval**

We acknowledge the importance of ethical considerations, including obtaining informed consent and adhering to Institutional Review Board (IRB) protocols, when handling human-related data.

We have obtained IRB approval from our organization for the collection and use of this dataset. Specifically, informed consent was obtained from every researcher affiliated with the Australian National University who participated in the study. By adhering to these rigorous ethical standards, we aim to protect the rights and privacy of all individuals involved and to maintain the integrity of our research.

**3. Copyright**

We intend to demonstrate that the videos we collected from online platforms fall under fair use. Fair use is a legal doctrine that says use of copyright-protected material under certain circumstances is allowed without permission from the copyright holder. [8] The copyright website of YouTube shows that: “Different countries/regions have different rules about when it’s OK to use material without the copyright owner’s permission. For example, in the United States, works of commentary, criticism, research, teaching, or news reporting might be considered fair use.” [9]. Similarly, the BiliBili and ItemFix have the same regulations for downloading the videos for fair use. Due to the fact that the primary authors are Australians, we also checked the Fair dealing provision in Australia Copyright Act 1968 [10]. Under section 103C(1) copyright in an audio-visual item, or in a work included in an audio-visual item, is not infringed by a fair dealing made for the purpose of research or study [11].

In our collection process, most videos are collected from YouTube (717 videos), and only a few videos are collected from BiliBili and ItemFix (3 videos). We ensured that all the videos were selected and used in accordance with the Fair Dealing provisions of the Australian Copyright Act 1968. Our approach was designed to adhere strictly to these legal frameworks, ensuring that the use of copyrighted material was only limited to research purposes. This careful consideration helps to protect the rights of the original content creators while allowing us to utilize the material in a legally compliant manner.

For the concerns of depreciated videos, we will regularly check the sources of our dataset to ensure that the videos or data links are still valid. This can involve automated scripts that periodically verify online links and alert you if any sources become inaccessible. If the original video source becomes unavailable, we will seek alternative sources or mirrors that host the same content. Keeping a list of alternative sources can help maintain the dataset's integrity.However, if no alternative sources are found, we may consider removing the unavailable content from our dataset to ensure that it remains up-to-date and reliable. Additionally, we will keep updating the experimental results on our dataset website if some videos are unavailable. We will also note the missing videos in the list for transparency and fairness comparison for future research.

---

### Author Rebuttal · Authors · 2024-08-17

**To all reviewers:**

**4. Bias**

To ensure diversity in our dataset, we have extensively collected videos from various regions and countries. The purpose of collecting videos from diverse regions and countries is to advance anomaly detection technologies globally, benefiting communities worldwide.
We conducted searches on platforms such as YouTube using text queries that combine various anomaly types (e.g., fighting, fire, people falling) with diverse region and country-specific terms. To maximize the diversity of videos retrieved from around the world, we also utilized queries in multiple languages using translators.

The collected videos represent a broad spectrum of ages, genders, ethnicities, and scenarios, which helps to reduce potential biases.

By incorporating such a diverse array of content, our goal is to create a balanced and representative dataset that accurately mirrors the complexity and diversity of real-world situations.


**Reference:**

[1] Download request online form for our MSAD dataset https://forms.microsoft.com/pages/responsepage.aspx?id=XHJ941yrJEaa5fBTPkhkN0_bcDHlPvFAiLdm3BQe86NURVI5RlRWODhYWVZYSzNCSlBROThBTEQzOC4u

[2] Terms and conditions of MSAD dataset: https://msad-dataset.github.io/agreement.html

[3] The GitHub repository deface: https://github.com/ORB-HD/deface

[4] Tian, Yu, et al. "Weakly-supervised video anomaly detection with robust temporal feature magnitude learning." Proceedings of the IEEE/CVF international conference on computer vision. 2021.

[5] Chen, Yingxian, et al. "Mgfn: Magnitude-contrastive glance-and-focus network for weakly-supervised video anomaly detection." Proceedings of the AAAI conference on artificial intelligence. Vol. 37. No. 1. 2023.

[6] Chen, Weiling, et al. "TEVAD: Improved video anomaly detection with captions." Proceedings of the IEEE/CVF Conference on Computer Vision and Pattern Recognition. 2023.

[7] Wu, Peng, et al. "Vadclip: Adapting vision-language models for weakly supervised video anomaly detection." Proceedings of the AAAI Conference on Artificial Intelligence. Vol. 38. No. 6. 2024.

[8] Fair use on YouTube:
https://support.google.com/youtube/answer/9783148?hl=en&ref_topic=2778546&sjid=11694239482035441950-AP

[9] Frequently asked questions about fair use:
https://support.google.com/youtube/answer/6396261?hl=en&ref_topic=2778546&sjid=11694239482035441950-AP

[10] Fair dealing provision in Australia:
https://anulib.anu.edu.au/research-learn/copyright/overview/fair-dealing

[11] COPYRIGHT ACT 1968 - SECT 103C
http://www8.austlii.edu.au/cgi-bin/viewdoc/au/legis/cth/consol_act/ca1968133/s103c.html

[12] MSAD original dataset submission link for review (Google Drive): https://drive.google.com/drive/folders/1mxFGCcAuEecN0c7MHD12wD4pO2Ej5kLe

---

### Decision · Program_Chairs · 2024-09-26

**Decision:**

Accept (Poster)

**Comment:**

The idea of providing diversity in new datasets based on the analysis of shortcomings in existing datasets is highly appreciated. While the suggested benchmark algorithm could have been enhanced by incorporating additional methods to increase its impact, it is still considered a solid starting point for others. The authors' response to the reviewers' critiques in the rebuttal, such as their review of related work, is also appreciated.